# SALoM: Structure Aware Temporal Graph Networks with Long-Short Memory Updater

**Hanwen Liu**
Zhejiang University
Hangzhou,Zhejiang
22451315@zju.edu.cn

**Longjiao Zhang**
Zhejiang University
Hangzhou,Zhejiang
zhljJoan@zju.edu.cn

**Rui Wang**[*]
Zhejiang University
High-Tech Zone (Binjiang) Institute
of Blockchain and Data Security
Hangzhou,Zhejiang
rwang21@zju.edu.cn

**Tongya Zheng**
Zhejiang Provincial Engineering Research Center
for Real-Time SmartTech in Urban Security Governance,
School of Computer and Computing Science,
Hangzhou City University
doujiang_zheng@163.com

**Sai Wu**
Zhejiang University
Hangzhou,Zhejiang
wusai@zju.edu.cn

**Chang Yao**
Zhejiang University
Hangzhou,Zhejiang
changy@zju.edu.cn

**Mingli Song**
Zhejiang University
Hangzhou,Zhejiang
brooksong@zju.edu.cn

## Abstract

Dynamic graph learning is crucial for accurately modeling complex systems by integrating topological structure and temporal information within graphs. While memory-based methods are commonly used and excel at capturing short-range temporal correlations, they struggle with modeling long-range dependencies, harmonizing long-range and short-range correlations, and integrating structural information effectively. To address these challenges, we present SALoM: Structure Aware Temporal Graph Networks with Long-Short Memory Updater. SALoM features a memory module that addresses gradient vanishing and information forgetting, enabling the capture of long-term dependencies across various time scales. Additionally, SALoM utilizes a long-short memory updater (LSMU) to dynamically balance long-range and short-range temporal correlations, preventing over-generalization. By integrating co-occurrence encoding and LSMU through information bottleneck-based fusion, SALoM effectively captures both the structural and temporal information within graphs. Experimental results across various graph datasets demonstrate SALoM's superior performance, achieving state-of-the-art results in dynamic graph link prediction. Our code is openly accessible at https://github.com/wave5418/SALoM.

## 1 Introduction

Dynamic graphs [35] are essential for modeling intricate systems such as traffic planning [33], genomics [15], financial analysis [22], and environmental science [2]. Continuous-time dynamic graph learning networks [6, 17, 25] operate in a continuous stream of events, enabling the understanding of entity interactions. These networks excel in analyzing and predicting data patterns by incorporating

---

[*]Rui Wang is the corresponding author

39th Conference on Neural Information Processing Systems (NeurIPS 2025).

*temporal information* and *topological structure* [9, 31], making them valuable in social network analysis [26], fraud detection, recommendation systems [10], and predictive maintenance [19], etc.

To capture **temporal correlation information** between nodes over time, two common approaches are used: memory-based methods and neighbor-based sequence methods. *Memory-based methods*, like JODIE [17], DyRep [28], and TGN [25], assign vector-based memory data to each node to represent historical interaction sequences. These memory data are continuously updated by new events using RNN architectures, preserving temporal information related to all events associated with the respective nodes. On the other hand, *neighbor-based sequence methods* like DyGFromer [34] and CNEN [7] leverage sequential models such as attention mechanisms and transformer encoders. These methods integrate historical event lists with features from neighboring nodes and time intervals to effectively capture temporal correlations from neighbor sequences.

To characterize **topological structural features** of a subgraph formed by each node and its historical neighbors, various approaches are commonly used, including message passing, random walk, and specialized structural encodings. *Message passing*, a traditional method in graph neural networks like TGN [25], encodes the topological structure of the graph into node representations by iteratively aggregating information from neighboring nodes at each layer. *Random walk-based methods*, such as CAWN [31], generate random paths in the graph to extract information and contextual relationships from neighboring nodes. *Specialized structural encodings* like path counting [27] and co-occurrence neighbor encoding [34] enhance structural features and tackle issues like oversmoothing [18] and over-squashing [1]. In particular, co-occurrence neighbor encoding computes the co-occurrence of historical neighbors, providing notable advantages in various tasks.

Despite progress in the field, current methods still faces challenges in effectively capturing both long sequence temporal information and graph structure features, hindering model performance. Memory-based techniques *struggle with capturing complete long-range neighborhood temporal correlations*, due to issues like vanishing gradient and information forgetting in RNN architectures [8]. Neighbor-based sequential methods, on the other hand, *face difficulties in efficiently balancing long-term and short-term neighborhood features*. Moreover, *integrating long-term temporal and topological structural features poses a challenge*, given the issues of over-squashing and difficulty in node differentiation [7]. While co-occurrence neighbor encoding offers a potential solution, directly integrating it with temporal features may lead to conflicts and reduced performance.

To address the above challenges, we introduce Structure Aware temporal graph networks with Long-Short Memory updater(**SALoM**), a dynamic graph learning framework aimed at capture long-range temporal features, harmonize long-range and short-range dependencies, and integrating structural features within graphs. SALoM enables the model to discern subtle differences among isomorphic nodes and preserve critical temporal patterns. Our contributions can be summarized as follows:

- To address the challenge of capturing complete long-range neighborhood temporal correlations, we introduce a memory module that mitigates the gradient vanishing and information forgetting issues and effectively captures dynamic features across various time scales.

- Expanding on this memory module, we propose Long-Short Memory Updater (LSMU) to balance long-range and short-range temporal dependencies. LSMU utilizes a sparse mixture-of-experts module to integrate memory and encoded interactions for gate calculations, effectively adapting both influences and mitigating over-generalization issue.

- To tackle the lack of structure information, we propose to incorporate co-occurrence encoding into LSMU via information bottleneck-based fusion, effectively capturing and adaptively balancing the impact of temporal and structure encoding.

- We implement the prototype of **SALoM** and conduct extensive experiments to demonstrate its effectiveness. Our results show that SALoM outperforms the state-of-the-art dynamic graph learning methods on most benchmark datasets for link prediction. In particular, SALoM demonstrates significant improvements in prediction accuracy on the USLegis, UNtrade, and UNvote datasets where previous methods fell short, with enhancements of 14.18%, 15.89%, and 32.48% over its closest competitors, respectively.

## 2 Background and Related Works

### 2.1 Temporal Correlation Encoding in Dynamic Graph

**Memory-based Methods.** Memory-based methods use specialized memory modules to update evolving node representations through sequential interactions, theoretically preserving complete historical patterns. Jodie [17] pioneered this with RNNs and t-Batch, excelling in recommendation systems but lacking generalizability on other tasks. DyRep [28] extended memory-based methods to general dynamic graphs, emphasizing the interaction between graph structure and temporal dynamics. Innovations like TGAT's [32] attention mechanisms and Temporal Graph Networks (TGN)[25] combine memory modules with Multi-Head Attention, achieving early state-of-the-art results. However, memory updated iteratively based on RNN face practical limitations, such as gradient vanishing and information forgetting, which hinder long-range dependencies. And implementing discrete models on irregularly-sampled events will cause intra-batch information loss.

**Neighbor-Based Sequential Methods.** Some recent studies[7, 34] abandon iterative memory updates and directly process long historical neighbor feature sequences through sequential models, such as linear layers and transformer layers, to extract temporal correlations and structural features. While circumventing RNN-related gradient issues, information forgetting, and intra-batch loss, this approach imposes theoretical constraints on long-range dependency modeling through finite neighbor sampling and incurs memory overheads that scale quadratically with sequence length (e.g., $\mathcal{O}(L^2)$ for attention-based models), posing computational bottlenecks in practical applications.

### 2.2 Structural Encoding in Dynamic Graphs

**Message Passing Methods.** These methods leverage neighborhood aggregation without explicit structural encoding, implicitly capturing local topology through aggregating information from neighbors and propagating it layer by layer[25, 32]. These methods are often combined with memory-based techniques, such as GCN [16] and MHA [29], which exemplifies the application of this approach.

**Random Walk-Based Methods**. Inspired by static GNNs, early efforts to explicitly capture structure encoding in continuous-time dynamic graphs explored methods based on random walks. For instance, CAWN[31] extracts multiple causal anonymous walks for each node and employs RNNs to encode these walks and aggregates them to form the final node representation, with a particular emphasis on capturing causality in dynamic graphs.

**Specialized Structural Encodings**. As research advanced, various explicit structure encoding methods were developed[27, 31, 34], with co-occurrence neighbor encoding proving to be the most expressive and generalized. This approach, serving as a relative structure encoding, measures the frequency of common neighbors in the historical neighbor lists of two nodes. When integrated with temporal and node-specific features and processed by sequential models such as Transformers, it demonstrates significant improvements on continuous-time dynamic graph tasks, showing robust expressiveness and generalizability across diverse datasets.

### 2.3 Limitations of Existing Methods

Although some achievements have been made, the existing methods still have limitations in capturing long sequence information and graph structure information.

**Limitation#1: Difficulty capturing complete long-range neighborhood temporal correlations.** Memory-based methods, although theoretically capable of storing information from all historical interactions in memory data, struggle to effectively handle long-term dependencies in practice. This is because RNN architectures used for capturing temporal correlations and updating memory are susceptible to issues like vanishing gradient [8, 11, 14] and information forgetting [8, 11, 14]. Neighbor-based sequential methods, on the other hand, can capture long-range temporal correlations of historical interactions by sampling a long neighbor list. However, this approach leads to redundant computational overhead due to duplicated interactions related to the same node in the long neighbor lists, and it faces limitations on the interaction history due to constraints on neighbor list length. For a detailed motivation study of long-range temporal correlations capture, refer to Appendix B.1.

**Limitation#2: Challenge in balancing long-range and short-range temporal neighborhood features.** In dynamic graph learning, it is crucial to capture both long-term trends and short-term

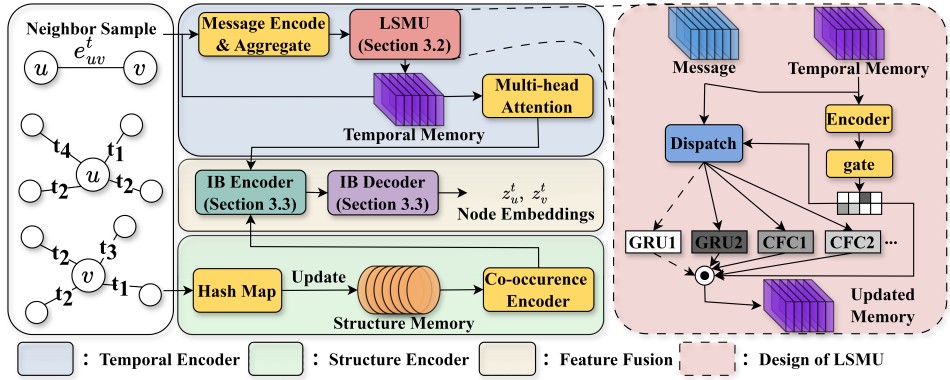

Figure 1: Model overview of SALoM

changes in neighborhood feature data, because long-term memory helps identify evolving trends and persistent patterns, while short-term events reflect local dynamics and immediate changes[12, 36]. However, existing memory-based methods struggle to adaptively balance the influence of different historical interactions across different time periods. Although neighbor-based sequence methods can assign uniform weights for historical events with different time periods[23], their complexity grows quadratically with the neighbor list length. They also struggle to adaptively determine the importance of short-range versus long-range correlations. Detailed motivation study of short-range temporal correlations capture and long-short term temporal correlation fusion refer to Appendix B.2 and B.3.

**Limitation#3: Hard to integrate long-term temporal and topological structural features.** For topological structural features, traditional message passing methods struggle to effectively handle long-term neighbors, often resulting in over-squashing and the inability to differentiate between homogeneous nodes. This is due to the over compression of distant node information, leading to a loss of crucial structural features[3, 24]. As a result, these methods primarily focus on local neighborhoods. While random walk-based methods can capture long-range dependencies, they can be time-consuming or limited by cached historical neighbors, hindering the utilization of long-range information. Co-occurrence neighbor coding offers a solution but is restricted by the reliance on neighbor lists, limiting its ability to incorporate long-term information. Additionally, directly fusing temporal and topological structural features may result in conflicts and ultimately degrade performance. For details on structure encoding and feature fusion, see Appendix B.4 and B.5.

## 3 Methodology

The SALoM framework is a cutting-edge approach designed to tackle key challenges within memory-based architectures, achieving three primary objectives that distinguish it from existing methods. An overview of SALoM is provided in Figure 1. SALoM excels at capturing long-range temporal dependencies through its **Continuous-Time Memory Module**, which uses ordinary differential equations to model continuous memory. The module effectively identifies long-term patterns in highly connected nodes, avoiding the discontinuity issues typical of batch-updated memory units. Additionally, SALoM features the **Long-Short Memory Updater(LSMU)**, which combines long-range dependencies from the Continuous-Time Memory Module with short-range dependencies from RNNs. By adjusting its focus based on input characteristics, the model becomes more responsive to changes, improving prediction accuracy. SALoM also enhances structural information using **HashMap Memory** and **Co-occurrence Encoder**. It integrates temporal dependencies and structural features through the **IB Encoder and Decoder**, resolving potential coding conflicts and reducing the over-squeezing problem found in traditional methods. This allows for better differentiation between similar nodes, boosting the model's representational power and understanding of data relationships.

### 3.1 Continuous-Time Memory Module

**Memory Update Based on Neural Ordinary Differential Equations.** We propose a continuous-time memory module based on Neural Ordinary Differential Equations (ODEs) to capture long-range temporal dependencies in nodes with extensive neighborhoods. Unlike discrete models, Neural ODEs

describe the dynamics of hidden states using differential equations, allowing for continuous memory evolution that aligns with irregular temporal interactions in real-world graphs.

The continuous time graph, which is modeled as a sequence of time-stamped events, is denoted as $\mathcal{G} = \{(u_1, v_1, t_1), (u_2, v_2, t_2), \cdots, (u_n, v_n, t_n)\}$, representing the addition or change of a node or interaction between a pair of nodes at times $0 \leq t_1 \leq t_2 \leq \cdots \leq t_n$. An interaction event between nodes $u$ and $v$ at timestamp $t$ is associated with temporal edge feature $e_{uv}^t \in \mathbb{R}^{d_E}$. Each node $u \in \mathcal{N}$ at timestamp $t$ is associated with raw feature $x_u^t \in \mathbb{R}^{d_N}$. Temporal memory of node $i$ at time $t$ before update, denoted as $M_{tem}^{t-}(i)$, is updated with message $msg_i^t$ with dimension $d_{msg}$ to $M_{tem}^t(i)$. We treat node memory updates as a continuous flow, expressed as:

$$\frac{dM_{tem}^t(i)}{dt} = J(M_{tem}^t(i), t, \theta), \tag{1}$$

where $\theta$ represents the training parameters. The function $J(\cdot)$ captures both the node's dynamics and the influence of neighboring nodes over time, mitigating the loss of temporal continuity from long-range batched updates. This approach effectively captures long-range dependencies in complex graphs by modeling the derivative of the target function rather than a direct input-output mapping, making it ideal for understanding temporal relationships.

**Efficient ODE Solvers.** However, the high computational overhead of ODE solvers presents a significant barrier [4, 13]. To address this issue, we explored a closed-form solution variant of Neural ODEs [13] for memory updates, which is quoted as CFC Cell in the rest of the paper. The memory update can be formulated as:

$$\begin{aligned} M_{tem}^t(i) =&\sigma(-f(M_{tem}^{t-}(i), msg_i^t, \theta_f) \cdot t) \odot g(M_{tem}^{t-}(i), msg_i^t, \theta_g) \\ &+ [1 - \sigma(-f(M_{tem}^{t-}(i), msg_i^t, \theta_f) \cdot t)] \odot h(M_{tem}^{t-}(i), msg_i^t, \theta_h), \end{aligned} \tag{2}$$

Here, $\sigma$ is the sigmoid activation function. The $\theta_f$, $\theta_g$, and $\theta_h$ are trainable model parameters. Denote $M_{tem}^{t-}(i)$ and $msg_i^t$ as the memory of node $i$ before timestamp $t$ and the aggregated message of node $i$ at time $t$, respectively. The function $f(M_{tem}^{t-}(i), msg_i^t, \theta_f)$ serves as the liquid time constant for the sigmoidal time gates, while $g(M_{tem}^{t-}(i), msg_i^t, \theta_g)$ and $h(M_{tem}^{t-}(i), msg_i^t, \theta_h)$ construct the potential memory of node $i$ at time $t$. These functions are instances of neural networks, such as multilayer perceptrons(MLPs). $\odot$ represents the Hadamard product. This formulation allows for efficient memory updates while maintaining the advantages of continuous-time modeling.

## 3.2 Long-Short Memory Updater as Temporal Memory Updater

**Bottleneck in only long-range dependencies.** While continuous-time memory modules effectively capture long-range temporal evolution, they can underperform on certain datasets, as illustrated in Figure 3. For instance, replacing the memory updater with a continuous-time module on the USLegis dataset led to decreased performance. This decline arises because interactions in USLegis are more influenced by recent discrete events than by long-term patterns. Overemphasizing long-range dependencies can cause node representations to overly generalize, neglecting recent events. Thus, we incorporate methods for both long- and short-range dependencies to balance long-range correlations with short-term influences.

**Adaptive Memory Backbone Selection.** The optimal balance between long-range and short-range temporal dependencies exhibits dataset-specific and entity-specific characteristics in continuous-time dynamic graphs. To address this heterogeneity, we implement a Sparse Mixture-of-Experts (MoE) framework [5] with adaptive expert selection based on temporal interaction patterns. As shown in Figure 1, our architecture integrates three complementary components - CFC Cells for full-capacity long-term dependency modeling, GRU Cells for robust short-term pattern capture, and Sparse MoE Controller for context-aware backbone selection. SALoM dynamically routes input samples using both temporal messages and node memory states. For each interaction event $(u, v, t)$, we generate time-aware messages through:

$$msg_u^t = \text{Linear}_{d_{smg}}(\text{concat}(M_{tem}^{t-}(u), M_{tem}^{t-}(v), \text{TE}(\Delta t), e_{uv}^t)), \tag{3}$$

where $\text{TE}(\cdot)$ denotes the temporal encoder [25] and $e_{uv}^t$ is the raw edge feature. The message $msg_v^t$ follows symmetric generation. Let $U_q(\cdot)$ denote the $q$-th expert network. The memory update

combines expert outputs through gated aggregation:

$$M_{tem}^t(u) = \sum_{q=1}^{Q} w_u^t(q) \cdot U_q\left(\text{msg}_u^t, M_{tem}^{t-}(u)\right). \tag{4}$$

Here, $Q$ is the total number of experts. Assume that $r$ experts are activated for learning. The routing weights $w_u^t$ are determined by:

$$\text{Score}_u^t = \text{MLP}\left(\text{concat}(\text{msg}_u^t, M_{tem}^{t-}(u))\right) \cdot A, \tag{5}$$

$$\text{Score}_u^t(q) = \begin{cases} \text{Score}_u^t(q) & \text{if } q \in \text{top-}r(\text{Score}_u^t) \wedge \text{Score}_u^t(q) > 0, \\ 0 & \text{otherwise,} \end{cases} \tag{6}$$

$$w_u^t = \text{Softmax}(\text{Score}_u^t). \tag{7}$$

where $A$ is a trainable matrix that computes raw scores for each expert based on mixed messages. The top-$r$ selects the $r$ experts with the highest scores. Then, we use a one-layer multi-head attention mechanism to aggregate neighbor memory, forming the temporal encoding of node $u$. The neighbors of $u$ are represented by $(u, u', t') \in \mathcal{G}$ for $t' < t$, with $x_u^t$ as the feature of node $u$ at time $t$ and $\phi(\cdot)$ as the time encoding [32]. The temporal encoding can be formulated as follows:

$$z_{tem}^t(u) = \text{Linear}_d(\text{MHA}(Q = (x_u^t + M_{tem}^t(u))\|\phi(0), K = K_u^t, V = V_u^t)), \tag{8}$$

$$K_u^t = V_u^t = \text{stack}(\{M_{tem}^{t-}(u')\|e_{uu'}^{t'}\|\phi(t - t') \mid (u, u', t') \in \mathcal{G}\}, \text{dim} = 0). \tag{9}$$

### 3.3 Combining Structural and Temporal Insights in Feature Fusion

To improve topological structural features while preserving long-term temporal characteristics, we present a new learning architecture that combines long-term temporal memory with specific structural encoding, known as the Co-occurrence Neighbor Encoder.

**Co-occurrence Neighbor Encoder as Structure Enhance Encoder.** We discuss how to capture structure features with co-occurrence neighbor encoding. Let the $k$-hop neighbors of node $u$ before timestamp $t$ be denoted as $S_k^t(u) = \{u' \mid (u, u', t') \in \mathcal{G}, t' < t\}$. We maintain two structural memory arrays for each node $u$, $M_{\text{struc}}^l[u, \cdot]$ and $M_{\text{struc}}^s[u, \cdot]$, which are used to store long-range and short-range historical neighbor hash tables, respectively. The lengths of these units correspond to the dimensions of the long-term memory unit $d_{s,l}$ and the short-term memory unit $d_{s,s}$, where $d_{s,l} = 4 \cdot d_{s,s}$. The long-range co-occurrence neighbor count of node $u$ with respect to $i$ can be formulated as:

$$O_{k,l}^t(u, i) = \sum_{j=0}^{d_{s,l}-1} \mathbb{I}(M_{struc}^l[u, j] = M_{struc}^l[i, j]), \tag{10}$$

which quantifies the number of common nodes between $u$ and $i$ in their hash memory. Similarly, the short-range co-occurrence neighbor count of node $u$ with respect to $i$ can be estimated using a similar approach. The structural encoding of node $u$ is derived from the structure memory as:

$$C_k^t(u) = \{O_{k,l}^t(u, u'), O_{k,l}^t(v, u'), O_{k,s}^t(u, u'), O_{k,s}^t(v, u') \mid u' \in S_k^t(u)\}, \tag{11}$$

$$z_{struc}^t(u) = \text{FFN}(C_k^t(u)), \tag{12}$$

where $\text{FFN}(\cdot)$ is a Feed-Forward Network. Appendix C.1 details structure memory updating process.

**Information Bottleneck Based Feature Fusion.** We employ an information bottleneck-based feature fusion approach to integrate temporal and structural encodings, as simple concatenation fails to address modality conflicts. Figure 3 highlights the performance decline from direct concatenation. This fusion is achieved using the IB Encoder and IB Decoder architectures.

Denote the unified node embedding of node $u$ at time $t$ as $z_u^t$. Temporal and structure encoding are denoted as $z_{tem}^t(u)$ and $z_{struc}^t(u)$ respectively. The label of node $u$ at timestamp $t$ is denoted as $y_u^t$.

IB Encoder obtains a hybrid encoding of structural and temporal information through a linear layer, and then applies variational approximation with a standard normal distribution to derive the representation of the unified node embedding $z_u^t$. The specific computation proceeds as follows.

$$\widetilde{z}_u^t = \text{Linear}_d(\text{concat}(z_{tem}^t(u), z_{struc}^t(u))), \tag{13}$$

$$\mu_u^t = \text{Linear}_d(\widetilde{z}_u^t), \tag{14}$$

$$\sigma_u^t = \log(1 + e^{(\mu_u^t - 5)}). \tag{15}$$

The IB Decoder generates the final representation of the unified node embedding based on the approximate distribution, denoted as $z_u^t \sim \mathcal{N}(\mu_u^t, \sigma_u^t)$.

$$z_u^t = \text{randn}(\mu = \mu_u^t, \sigma = \sigma_u^t). \tag{16}$$

The optimization objective and loss function are designed to maximize the information corresponding to $y_u^t$ within $\text{concat}(z_{tem}^t(u), z_{struc}^t(u))$, also denoted as $z_{concat}^t(u)$. We opt to minimize the mutual information between $z_{concat}^t(u)$ and $z_u^t$, which can measure the correlation between two variables.

$$\underset{z_u^t}{\arg\min} -I(z_u^t, y_u^t) + \beta \cdot I(z_u^t, z_{concat}^t(u)). \tag{17}$$

In this expression, $-I(z_u^t, y_u^t)$ represents the mutual information between the intermediate representation $z_u^t$ and the labels $y_u^t$, while $I(z_u^t, z_{concat}^t(u))$ represents the mutual information between the intermediate representation and the original representation. The former is aimed at maintaining representation relevant to the labels, the latter is aimed at denoising the representation. This leads to intermediate features that are easily distinguishable, preserving the original information while enhancing predictive power.

As mutual information is difficult to calculate, by employing variational approximation, we scale the target function to derive the mathematical form of its upper bound. This upper bound can then be transformed into a form combining Binary Cross-Entropy (BCE) loss and KL divergence. Due to space constraints, the detailed mathematical derivation for solving the upper bound of the optimization objective is provided in the appendix C.2.

$$-I(z_u^t, y_u^t) + \beta \cdot I(z_u^t, z_{concat}^t(u)) \leq y \cdot \log(p(y_u^t)) + (1 - y_u^t) \cdot \log(1 - p(y_u^t)) + \beta \cdot \text{KL}[z_u^t, \mathcal{N}(0, 1)]. \tag{18}$$

Based on the upper bond of the original optimization objective, the loss function for each node exemplified by node $u$ is as follows, where $\hat{y}$ is the model prediction.

$$loss = \text{BCE}(\hat{y}_u^t, y_u^t) + \beta \cdot \text{KL}(z_u^t, \mathcal{N}(0, 1)). \tag{19}$$

## 4 Experiments

### 4.1 Experiment Settings

**Datasets and Baselines.** We evaluate on thirteen datasets (Wikipedia, Reddit, MOOC, LastFM, Enron, Social Evo., UCI, Flights, Can. Parl., US Legis., UN Trade, UN Vote, and Contact) obtained from Edgebank [21]. Following DyGLib [34], we compare **SALoM** against ten popular dynamic graph learning methods, including JODIE [17], DyRep [28], TGAT [32], TGN [25], CAWN [31], EdgeBank [21], TCL [30], GraphMixer [9], DyGFormer [34], and CNE-N [7]. The details of the above datasets and baselines are elaborated in Appendix A.1 and Appendix A.2, respectively.

**Evaluation Metrics.** We evaluate the performance on the dynamic link prediction task, which involves predicting the presence of a link at a given time. This task includes two settings: the transductive setting predicts future links among nodes observed during training, while the inductive setting assesses link prediction for unseen nodes. We measure performance using Average Precision (AP) and Area Under the Receiver Operating Characteristic Curve (AUC-ROC) as evaluation metrics.

**Model Configuration.** Our **SALoM** is built upon the classic memory-based model TGN. We set structure memory dimension as $d_{s,l} = 64$, $d_{s,l} = 16$ and temporal memory dimension as $d_t = 172$. We choose 2 experts from a total of 3 GRU units and 3 CFC units. For IB fusion, we set $\beta = 1\text{e}^{-3}$.

**Implementation Details.** To ensure a fair comparison, we evaluate the baseline model TGN, the state-of-the-art model CNE-N, and our **SALoM** on the same machine with identical settings. We conduct the evaluation on an Ubuntu machine featuring an Intel(R) Xeon(R) Platinum 8352V CPU @ 2.10GHz and an NVIDIA GeForce RTX 4090 GPU with 24 GB memory. The models are trained for 100 epochs with early stopping using a patience of 20. The model exhibiting the best performance on the validation set is chosen for testing. A uniform learning rate of 0.0001 is applied to all methods across all datasets. Batch sizes are set to 10 for memory-based methods and 200 for neighbor-based sequence methods. Each dataset is divided into training/validation/testing sets in a 70%/15%/15% ratio. We conduct five runs of each method with seeds ranging from 0 to 4 and report the average performance. For the remaining baselines, we refer to the reported best performance in DyGFormer to maintain consistency, following a similar approach as in CNE-N [7].

Table 1: AP&AUC-ROC (%) for transductive and inductive link prediction.

| Metrics | Datasets | JODIE | DyRep | TGAT | TGN | CAWN | EdgeBank | TCL | GraphMixer | NAT | DyGFormer | CNE-N | SALoM |
|---|---|---|---|---|---|---|---|---|---|---|---|---|---|
| Trans-AP | Wikipedia | 96.50 | 94.86 | 96.94 | 98.28 | 98.76 | 90.37 | 96.47 | 97.25 | 97.50 | **99.03** | 98.61 | **99.03** |
| | Reddit | 98.31 | 98.22 | 98.52 | 98.47 | 99.11 | 94.86 | 97.53 | 97.31 | 99.10 | 99.22 | 99.26 | **99.27** |
| | MOOC | 80.23 | 81.97 | 85.84 | **93.21** | 80.15 | 57.97 | 82.38 | 82.78 | 87.21 | 87.52 | 90.16 | 92.42 |
| | LastFM | 70.85 | 71.92 | 73.42 | 84.36 | 86.99 | 79.29 | 67.27 | 75.61 | 88.57 | 93.00 | 92.60 | **93.14** |
| | Enron | 84.77 | 82.38 | 71.12 | 91.51 | 89.56 | 83.53 | 79.70 | 82.25 | 90.81 | 92.47 | 92.13 | **94.08** |
| | Social Evo. | 89.89 | 88.87 | 93.16 | 89.83 | 84.96 | 74.95 | 93.13 | 93.37 | 91.23 | **94.73** | 94.50 | **94.73** |
| | UCI | 89.43 | 65.14 | 79.63 | 92.94 | 95.18 | 76.20 | 89.57 | 93.25 | 94.26 | 95.79 | 95.64 | **96.36** |
| | Flights | 95.60 | 95.29 | 94.03 | 97.94 | 98.51 | 89.35 | 91.23 | 90.99 | 97.66 | 98.91 | 98.73 | **98.94** |
| | Can. Parl. | 69.26 | 66.54 | 70.73 | 96.29 | 69.82 | 64.55 | 68.67 | 77.04 | 83.83 | 97.36 | 81.84 | **99.11** |
| | US Legis. | 75.05 | 75.34 | 68.52 | 78.09 | 70.58 | 58.39 | 69.59 | 70.74 | 77.56 | 71.11 | 72.58 | **92.27** |
| | UN Trade | 64.94 | 63.21 | 61.47 | 68.3 | 65.39 | 60.41 | 62.21 | 62.61 | 72.32 | 66.46 | 77.97 | **93.86** |
| | UN Vote | 63.91 | 62.81 | 52.21 | 64.13 | 52.84 | 58.49 | 51.90 | 52.11 | 69.70 | 55.55 | 58.10 | **86.81** |
| | Contact | 95.31 | 95.98 | 96.28 | 95.00 | 90.26 | 92.58 | 92.44 | 91.92 | 97.25 | 98.29 | 98.28 | **98.53** |
| | Avg. Rank | 7.76 | 8.61 | 8.38 | 4.92 | 7.07 | 10.53 | 9.76 | 8.38 | 4.53 | 3.15 | 3.61 | **1.07** |
| Trans-AUC | Wikipedia | 96.33 | 94.37 | 96.67 | 98.01 | 98.54 | 90.78 | 95.84 | 96.92 | 96.72 | **98.91** | 98.4 | 98.87 |
| | Reddit | 98.31 | 98.17 | 98.47 | 98.32 | 99.01 | 95.37 | 97.42 | 97.17 | 99.02 | 99.15 | 99.19 | **99.2** |
| | MOOC | 83.81 | 85.03 | 87.11 | **93.56** | 80.38 | 60.86 | 83.12 | 84.01 | 88.38 | 87.91 | 91.42 | 92.52 |
| | LastFM | 70.49 | 71.16 | 71.59 | 82.66 | 85.92 | 83.77 | 64.06 | 73.53 | 86.94 | **93.05** | 92.21 | 92.32 |
| | Enron | 87.96 | 84.89 | 68.89 | 90.99 | 90.45 | 87.05 | 75.74 | 84.38 | 92.02 | 93.33 | 92.77 | **95.11** |
| | Social Evo. | 92.05 | 90.76 | 94.76 | 90.36 | 87.34 | 81.60 | 94.84 | 95.23 | 93.22 | 96.3 | 96.20 | **96.36** |
| | UCI | 90.44 | 68.77 | 78.53 | 92.17 | 93.87 | 77.30 | 87.82 | 91.81 | 93.02 | 94.49 | 94.32 | **95.53** |
| | Flights | 96.21 | 95.95 | 94.13 | 97.99 | 98.45 | 90.23 | 91.21 | 91.13 | 97.32 | 98.93 | 98.74 | **98.99** |
| | Can. Parl. | 78.21 | 73.35 | 75.69 | 97.17 | 75.7 | 64.14 | 72.46 | 83.17 | 87.70 | 97.76 | 84.49 | **99.18** |
| | US Legis. | 82.85 | 82.28 | 75.84 | 84.63 | 77.16 | 62.57 | 76.27 | 76.96 | 84.68 | 77.90 | 79.38 | **93.75** |
| | UN Trade | 69.62 | 67.44 | 64.01 | 69.41 | 68.54 | 66.75 | 64.72 | 65.52 | 76.76 | 70.20 | 79.64 | **93.23** |
| | UN Vote | 68.53 | 67.18 | 52.83 | 62.76 | 53.09 | 62.97 | 51.88 | 52.46 | 74.44 | 57.12 | 60.67 | **87.87** |
| | Contact | 96.66 | 96.48 | 96.95 | 95.37 | 89.99 | 94.34 | 94.15 | 93.94 | 97.64 | 98.53 | 98.62 | **98.69** |
| | Avg. Rank | 7.07 | 8.38 | 8.69 | 5.53 | 7.15 | 10.15 | 10.07 | 8.61 | 4.30 | 3.23 | 3.53 | **1.23** |
| Ind-AP | Wikipedia | 94.82 | 92.43 | 96.22 | 97.49 | 98.24 | - | 96.22 | 96.65 | 95.40 | **98.59** | 97.76 | 98.49 |
| | Reddit | 96.5 | 96.09 | 97.09 | 97.26 | 98.62 | - | 94.09 | 95.26 | 98.56 | 98.84 | 98.82 | **98.93** |
| | MOOC | 79.63 | 81.07 | 85.50 | **91.86** | 81.42 | - | 80.60 | 81.41 | 83.59 | 86.96 | 88.71 | 90.53 |
| | LastFM | 81.61 | 83.02 | 78.63 | 87.18 | 89.42 | - | 73.53 | 82.11 | 86.87 | 94.23 | 94.00 | **94.56** |
| | Enron | 80.72 | 74.55 | 67.05 | 84.53 | 86.35 | - | 76.14 | 75.88 | 89.03 | 89.76 | 87.59 | **91.67** |
| | Social Evo. | 91.96 | 90.04 | 91.41 | 82.85 | 79.94 | - | 91.55 | 91.86 | 91.22 | **93.14** | 92.70 | 92.84 |
| | UCI | 79.86 | 57.48 | 79.54 | 82.04 | 92.73 | - | 87.36 | 91.19 | 87.30 | **94.54** | 93.58 | 94.36 |
| | Flights | 94.74 | 92.88 | 88.73 | 95.03 | 97.06 | - | 83.41 | 83.03 | 96.59 | 97.79 | 97.34 | **97.85** |
| | Can. Parl. | 53.92 | 54.02 | 55.18 | 78.75 | 55.80 | - | 54.30 | 55.91 | 60.62 | 87.74 | 65.01 | **96.20** |
| | US Legis. | 54.93 | 57.28 | 51.00 | 55.74 | 53.17 | - | 52.59 | 50.71 | 57.54 | 54.28 | 59.54 | **68.38** |
| | UN Trade | 59.65 | 57.02 | 61.03 | 77.86 | 65.24 | - | 62.21 | 62.17 | 69.29 | 64.55 | 69.84 | **85.46** |
| | UN Vote | 56.64 | 54.62 | 52.24 | 65.67 | 49.94 | - | 51.6 | 50.68 | 66.35 | 55.93 | 57.57 | 62.37 |
| | Contact | 94.34 | 92.18 | 95.87 | 88.56 | 89.55 | - | 91.11 | 90.59 | 96.79 | **98.03** | 97.58 | 97.79 |
| | Avg. Rank | 7.92 | 8.69 | 8.15 | 5.38 | 6.38 | - | 8.53 | 8.15 | 5.07 | 2.84 | 3.23 | **1.53** |
| Ind-AUC | Wikipedia | 94.33 | 91.49 | 95.9 | 97.08 | 98.03 | - | 95.57 | 96.30 | 94.74 | **98.48** | 97.45 | 98.26 |
| | Reddit | 96.52 | 96.05 | 96.98 | 96.94 | 98.42 | - | 93.8 | 94.97 | 97.99 | 98.71 | 98.69 | **98.85** |
| | MOOC | 83.16 | 84.03 | 86.84 | **92.02** | 81.86 | - | 81.43 | 82.77 | 86.13 | 87.62 | 89.94 | 90.09 |
| | LastFM | 81.13 | 82.24 | 76.99 | 85.58 | 87.82 | - | 70.84 | 80.37 | 83.07 | **94.08** | 93.62 | 93.77 |
| | Enron | 81.96 | 76.34 | 64.63 | 83.58 | 87.02 | - | 72.33 | 76.51 | 89.92 | 90.69 | 88.24 | **92.57** |
| | Social Evo. | 93.70 | 91.18 | 93.41 | 82.04 | 84.73 | - | 93.71 | 94.09 | 92.11 | **95.29** | 94.99 | 95.03 |
| | UCI | 78.80 | 58.08 | 77.64 | 86.48 | 90.40 | - | 84.49 | 89.30 | 83.81 | **92.63** | 91.31 | 92.17 |
| | Flights | 95.21 | 93.56 | 88.64 | 95.92 | 96.86 | - | 82.48 | 82.27 | 96.36 | 97.80 | 97.20 | **97.95** |
| | Can. Parl. | 53.81 | 55.27 | 56.51 | 80.21 | 58.83 | - | 55.83 | 58.32 | 61.62 | 89.33 | 66.51 | **96.07** |
| | US Legis. | 58.12 | 61.07 | 48.27 | 58.87 | 51.49 | - | 50.43 | 47.20 | 62.85 | 53.21 | 60.10 | **65.56** |
| | UN Trade | 62.28 | 58.82 | 62.72 | 75.70 | 67.05 | - | 63.76 | 63.48 | 72.56 | 67.25 | 71.40 | **83.04** |
| | UN Vote | 58.13 | 55.13 | 51.83 | 61.64 | 48.34 | - | 50.51 | 50.04 | 66.26 | 56.73 | 58.85 | 62.44 |
| | Contact | 95.37 | 91.89 | 96.53 | 88.87 | 89.07 | - | 93.05 | 92.83 | 96.67 | **98.30** | 97.91 | 97.98 |
| | Avg. Rank | 7.76 | 8.53 | 8.07 | 5.46 | 6.53 | - | 8.76 | 8.15 | 5.00 | 2.69 | 3.46 | **1.53** |

## 4.2 Performance Study on Model Accuracy

In this section, we analyze the model accuracy in terms of the AP & AUC-ROC metrics for the ten baseline models and our SALoM in transductive and inductive dynamic link prediction tasks. The results are presented in Table 1, highlighting the best and second-best performances using bold and underlined fonts. Note that EdgeBank is evaluated only in the transductive setting and its results for the inductive setting are not included. Our SALoM consistently achieves top performance across most datasets, with average rankings ranging from 1.07 to 1.46 for different metrics. Specifically, notable improvements are observed in the USLegis, UNtrade, and UNvote datasets, with enhancements of 14.18%, 15.89%, and 32.48% over its closest competitors, respectively. This superior performance can be attributed to SALoM's ability to effectively capture both long-range and short-range temporal dependencies using LSMU, enabling adaptive retention or forgetting of temporal correlations. Additionally, the information bottleneck-based fusion in SALoM allows for structural and temporal awareness without conflict, enabling nuanced node distinctions from multiple perspectives. We provide the version with standard deviates in Appendix D.1.

## 4.3 Ablation Study

**Effectiveness of LSMU.** In this section, we evaluate SALoM using various temporal memory updaters, including the traditional GRU from TGN, CFC (§3.1), MoE-GRU (GRUs in MoE architecture),

and LSMU (§3.2). Results on tested AP are shown in Figure 2. LSMU consistently outperforms other updaters, with an improvement of 3-5% in AP compared to GRU. This superiority is due to GRU's limitations in capturing long-range temporal features due to vanishing gradients and forgetting information issues. CFC captures long-range features but struggles with short-range ones and over-generalization issue. Simply using the MoE method does not solve these issues. LSMU excels by dynamically determining retention or forgetting of long-range and short-range temporal correlations using a sparse MoE module. This adaptive approach effectively balances influences, mitigating over-generalization for superior performance. We further experimented the training expense and universality of LSMU in Appendix D.2 and D.3.

**Effectiveness of Feature Fusion.** In this section, we evaluate SALoM using various fusion methods for temporal and structural embedding, including w/o SE (without structure encoding), Concat, Linear, and our proposed IB method (§3.3), and show the results in Figure 3. Simple Concat and Linear fusion methods offer advantages for UCI, Can.Parl, and UNtrade datasets, but struggle with the USLegis dataset due to potential conflicts between temporal and structural embeddings. This highlights the need for effective fusion methods. On the other hand, our IB fusion method effectively balances the influence of temporal and structure encoding, consistently delivering superior performance.

### 4.4 Performance under different numbers of historical neighbors.

In this section, we analyze SALoM's performance with varying numbers of sampled historical neighbors aggregated per iteration, as shown in Figure 4. Since SALoM builds on memory-based methods that update node memory iteratively, it does not require processing a large number of historical neighbors each iteration. Having 10 historical neighbors per iteration is sufficient for SALoM to perform well. This is because our proposed LSMU effectively retains valuable long-range features in memory data and dynamically balances temporal dependencies across different time. Performance declines when the number of neighbors aggregated each iteration exceeds 100, likely due to over-smoothing. Excessive neighbor sampling increases similarity between node embeddings during aggregation, reducing their distinctiveness. Moreover, since the iterative memory update mechanism already preserves long-term information, excessive sampling of historical neighbors disproportionately weights long-term dependencies and may dilute crucial short-term patterns.

### 4.5 Trade-off Between Accuracy and Efficiency

In this section, we compare the time per epoch and AP of SALoM with baseline methods, as shown in Figure 5. SALoM is evaluated with different batch sizes, while baseline methods maintain their default settings from their respective papers. Larger batch sizes in SALoM allow for more parallel event computations, enhancing computational efficiency. However, this efficiency gain comes at the cost of intra-batch information loss, impacting training accuracy. This trade-off is consistent with other memory-based methods. Our results suggest that SALoM can outperform existing methods with a slight edge in performance at comparable computational costs. When computational constraints are relaxed, setting a small batch size significantly boosts SALoM's performance, largely surpassing existing approaches in dynamic graph learning. We further study the impact of varying batch sizes on accuracy in Appendix D.4.

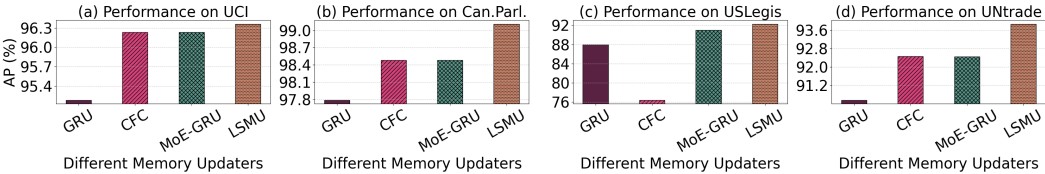

Figure 2: Ablation study on different memory updaters.

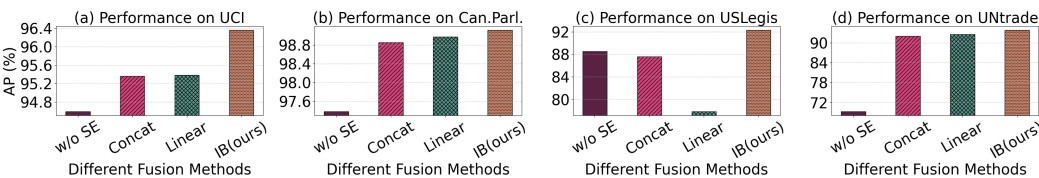

Figure 3: Ablation study on different fusion methods for structural and temporal embedding.

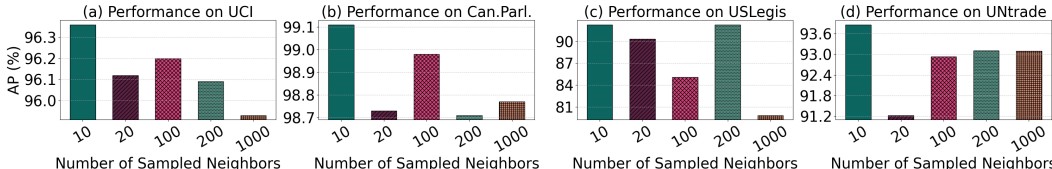

Figure 4: Ablation study on different numbers of historical neighbors aggregated each iteration.

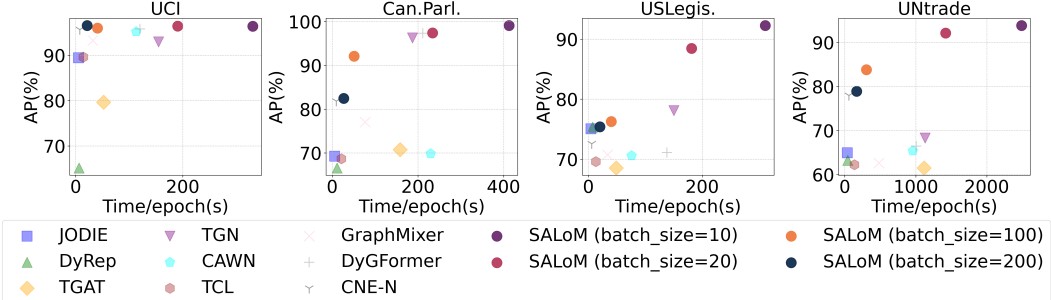

Figure 5: The trade-off between efficiency and performance under different settings of batch sizes.

Table 2: Leakage-free performance evaluation.

| Metrics | Methods | MOOC | UCI | Enron | Can. Parl. | US Legis. | UN Trade | UN Vote | Avg. Rank |
|---------|---------|------|-----|-------|-----------|-----------|----------|---------|-----------|
| | CAWN | 80.15 | 95.18 | 89.56 | 69.82 | 70.58 | 65.39 | 52.84 | 3.57 |
| | TCL | 82.38 | 89.57 | 79.7 | 68.67 | 69.59 | 62.21 | 51.9 | 4.86 |
| Trans-AP | GraphMixer | 82.78 | 93.25 | 82.25 | 77.04 | 70.74 | 62.61 | 52.11 | 3.57 |
| | DyGFormer | 87.52 | 95.79 | 92.47 | 97.36 | 71.11 | 66.46 | 55.55 | 2 |
| | SALoM | **91.39** | **96.39** | **93.19** | **98.79** | **75.79** | **92.52** | **68.52** | **1** |

## 4.6 Leakage-Free Evaluation

This section clarifies the information leakage issue in TGNN evaluation and presents our solution. The problem arises because: during batch training, edges sharing identical timestamps might be split across consecutive batches. When processing the second batch, the model's memory has already been updated by edges from the first batch that have the same timestamp, thus gaining access to information that should be chronologically unavailable, constituting an information leakage problem.

To ensure a rigorous leakage-free evaluation, we implement a time-aware dual memory management system. This approach maintains two separate memory states: one storing the final node state from the previous timestamp, and another for accumulating updates within the current timestamp. During inference for a given edge, the model strictly uses the memory state from timestamps prior to the current edge's time to form node representations. Only when advancing to the next distinct timestamp are the accumulated updates synchronized, preventing any leakage within the same timestamp.

As shown in Table 2, SALoM maintains state-of-the-art performance under these strict leakage-free conditions. While the impact of leakage is minimal on most datasets, performance variations are observed in USLegis (90%→75%) and UNvote (80%→68%). Crucially, SALoM still outperforms its best competitors by significant margins (4.68%and 12.97%in average precision, respectively), confirming its robust SOTA status through a fair and rigorous comparison. Further detailed examples and more experimental results are provided in Appendix C.3

## 5 Conclusion and Future Work

This paper introduces a continuous-time dynamic graph learning framework that emphasizes capturing temporal correlations and structural relations in graphs. We propose the Long-Short Memory Updater (LSMU) to extract both long-range and short-range temporal dependencies and balance their influence to mitigate over-globalization. By integrating o-occurrence encoding into LSMU through information bottleneck-based fusion, we unify temporal and structural information, improving model performance and achieving state-of-the-art results on benchmark datasets. Our experiments adhere to the DyGLib framework, enabling reproducibility and comparison with other methods. In the future, potential areas for improvement in our framework include exploring automatic feature extraction methods, enhancing neighbor aggregation efficiency, and improving strategies for mitigating intra-batch information loss.

## Acknowledgments

This research is supported by the "Pioneer" R&D Program of Zhejiang (No.2024C01019), the Zhejiang Province "Jianbing" Key R&D Project of China (No.2025C01010), the Hangzhou Joint Fund of the Zhejiang Provincial Natural Science Foundation of China (No.LHZSD24F020001), the Zhejiang Province High-Level Talents Special Support Program "Leading Talent of Technological Innovation of Ten-Thousands Talents Program" (No.2022R52046), and the Fundamental Research Funds for the Central Universities (No.2021FZZX001-23 and 226-2025-00067). The author gratefully acknowledges the support of Zhejiang University Education Foundation Qizhen Scholar Foundation.

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

# A   Datasets and Baselines

## A.1   Datasets Details

- **Wikipedia**: This dataset captures edits on Wikipedia pages over a one-month period. Nodes represent editors and wiki pages, while edges denote timestamped edit requests. Each edge is associated with LIWC feature vector derived from the edit text, encoding linguistic and psychological attributes.

- **Reddit**: Spanning one month, this dataset models interactions within Reddit subreddits. Nodes represent users or posts, and edges indicate timestamped posting requests. Similar to the Wikipedia dataset, edges are annotated with LIWC feature vectors based on the text of the posts.

- **MOOC**: This dataset represents a student interaction network within an online course. Nodes are students or content units (e.g., problem sets, videos), and edges reflect students accessing content units. Each edge has four features, capturing interaction-specific attributes.

- **LastFM**: This interaction network tracks 1,000 users listening to the 1,000 most popular songs over one month. Nodes represent users and songs, and edges indicate user-listens-to-song relationships. The dataset contains no additional edge or node attributes.

- **Enron**: This dataset comprises approximately 50,000 emails exchanged among Enron employees over three years. Nodes represent employees, and edges denote email correspondences. No attributes are included in this dataset.

- **Social Evo.**: This mobile phone proximity network tracks interactions in an undergraduate dormitory from October 2008 to May 2009. Nodes represent individuals, and edges indicate physical proximity, with each edge having two features describing the interaction.

- **UCI**: This dataset models a Facebook-like communication network among University of California, Irvine students. Edges represent timestamped interactions with second-level temporal granularity. The dataset includes no additional attributes.

- **Flights**: A directed dynamic network illustrating air traffic evolution during the COVID-19 pandemic. Nodes represent airports, and edges denote tracked flights, with edge weights indicating the number of flights between two airports per day. The dataset was specifically extracted and cleaned for this study.

- **Can. Parl.**: This dynamic political network documents interactions among Canadian Members of Parliament (MPs) from 2006 to 2019. Nodes represent MPs, and edges are formed when two MPs vote 'yes' on the same bill. Edge weights reflect the number of shared 'yes' votes in a year.

- **USLegis.**: This dataset captures co-sponsorship interactions among US Senate legislators. Nodes represent senators, and edge weights indicate the number of times two senators co-sponsored a bill in a given congressional session.

- **UN Trade**: A weighted, directed network of food and agriculture trade among 181 nations over 30 years. Nodes represent countries, and edge weights denote the normalized total value of agricultural imports or exports between pairs of countries.

- **UN Vote**: This dataset records roll-call votes in the United Nations General Assembly from 1946 to 2020. Nodes represent nations, and edges are formed when two nations vote 'yes' on the same item, with edge weights incremented by one per shared vote.

- **Contact**: This dataset tracks physical proximity among approximately 700 university students over four weeks. Nodes represent students with unique IDs, and edges indicate close physical proximity, with edge weights reflecting the strength of proximity.

## A.2   Baseline Details

- **JODIE**[17] targets bipartite networks with instantaneous user-item interactions. It utilizes two coupled recurrent neural networks (RNNs) to recursively update user and item representations. A projection operation is employed to predict the future representation trajectory of each user or item, enabling the model to capture evolving interaction patterns.

- **DyRep**[28] features a custom RNN to update node representations upon the observation of new edges. It incorporates a temporal attention mechanism, parameterized by the recurrent architecture, to assign weights to neighbors at each timestamp, effectively modeling temporal dependencies in node interactions.

- **TGAT**[32] aggregates features from a node's temporal-topological neighbors using a self-attention mechanism to compute node representations. It includes a time-encoding function to capture temporal patterns, enhancing its ability to model dynamic network structures.

- **TGN**[25] maintains evolving memory for each node, updating it through a message function, aggregator, and memory updater when a node participates in an interaction. An embedding module generates temporal node representations, balancing efficiency and expressiveness in dynamic settings.

- **CAWN**[31] extracts multiple causal anonymous walks for each node to explore network dynamics and generate relative node identities. It employs RNNs to encode these walks and aggregates them to form the final node representation, emphasizing causality in dynamic graphs.

- **EdgeBank**[21] is a memory-based approach for transductive dynamic link prediction. It stores observed interactions in a memory unit and updates it using various strategies. An interaction is predicted as positive if retained in memory and negative otherwise, offering a lightweight solution.

- **TCL**[30] generates node interaction sequences via a breadth-first search on temporal dependency interaction sub-graphs. It employs a graph transformer that integrates graph topology and temporal information, using cross-attention to model interdependencies between interacting nodes.

- **GraphMixer**[9] leverages a fixed-time encoding function, which outperforms trainable versions. It integrates this function into an MLP-Mixer-based link encoder to learn from temporal links, while a node encoder with neighbor mean-pooling summarizes node features.

- **NAT**[20] uses a dictionary-type neighborhood representation to aggregate temporal neighbors. It employs a recurrent process with random Fourier feature (RFF)-based time embedding to learn node representations, constructing query-induced subgraphs without neighbor sampling to reduce computational costs.

- **DyGFormer**[34] is a Transformer-based model that focuses on first-hop interactions between nodes. It introduces a neighbor co-occurrence encoding scheme via a patching method, feeding these into a Transformer to capture long-term correlations between source and destination node sequences.

- **CNE-N**[7] stores historical neighbor in hash-like tables, efficiently calculates co-occurrence encoding and combines other features to form node embeddings.

# B   Individual motivations

## B.1   Motivation of Continuous Models for Long-Term Temporal Correlation Capture

Traditional sequential methods (RNNs/GRUs) treat dynamic graphs as discrete events, struggling with temporal continuity and long-term dependencies due to gradient issues. In contrast, ODE-based methods preserve continuity and avoid gradient issues through second-order derivative integration, demonstrating superior long-term dependency capture with minimal information loss. However, our ablation studies in §4.3 show that over-emphasizing long-term dependencies can cause over-globalization. We therefore propose LSMU, an MoE-based approach that dynamically balances long/short-term dependencies, adaptively selecting the optimal processing.

## B.2   Motivation of GRU for Short-Term Temporal Correlation Capture

In dynamic graphs, recent neighbor interactions typically provide the most valuable information. While simple RNNs and K-neighbor aggregation fail to effectively capture these patterns (due to vanishing gradients and limited neighbor hops respectively), GRU's gating mechanism enables robust short-term dependency learning while complementing our ODE-based long-term capture.

Table 3: Motivation for short-term temporal correlation capture.

|  | CanParl | USLegis | UCI | Untrade |
|---|---|---|---|---|
| RNN | 98.81 | 89.90 | 96.00 | 92.82 |
| AggerateNeighbor | 98.77 | 77.31 | 96.22 | 92.83 |
| GRU | **99.11** | **92.27** | **96.36** | **93.86** |

Table 4: Motivation for sparse MoE as long-short term temporal correlation fuser.

|  | CanParl | USLegis | UCI | Untrade |
|---|---|---|---|---|
| Concat | 96.90 | 76.93 | 96.02 | 92.00 |
| Avg-Voting | 98.65 | 76.06 | 96.23 | 92.99 |
| MoE | **99.11** | **92.27** | **96.36** | **93.86** |

Our extended experiments confirm that simple RNN suffers from gradient vanishing, leading to performance degradation (mitigated by GRU's gating mechanism). Besides, K-neighbor aggregation is limited to 1–2 hops, as wider aggregation blurs node representations.

## B.3 Motivation of Sparse MoE for Long-Short Term Temporal Correlation Fusion

We propose MoE to fuse long-short term temporal correlations while adaptively selecting optimal backbones and weights per input. In temporal graphs, events exhibit mixed dependencies, but naive fusion methods (e.g., concatenation, voting) treat edges uniformly, ignoring their distinct evolutionary patterns. MoE addresses this by dynamically routing events to specialized experts based on node/edge features, achieving superior performance (see Table). For instance, on USLegis, MoE improves Trans.AP by 15.3% over concatenation and 16.2% over voting, demonstrating its ability to capture varied temporal scales.

This advantage stems from MoE's capability of adaptively fusing long-short term correlations. And learnable weights for feature fusion.

## B.4 Motivation for Co-Occurrence Encoding as Structure Encoding

We evaluated model performance with different structure encoding methods, comparing the expressivity of co-occurrence encoding and traditional methods, such as message passing and random-walk based methods.

As shown in Table 5 and Figure 5, co-occurrence encoding shows better expressivity compared with traditional methods. Improving model accuracy, especially 63.44%->93.86% on UNtrade.

Our co-occurrence encoder efficiently captures structural patterns while maintaining computational feasibility. Unlike traditional methods that trade off expressiveness for efficiency, it encodes neighbor co-occurrence frequencies as relative structural features, enhanced by a hash-based memory system for accelerated processing. Experiments demonstrate its superiority over basic message passing and random walk approaches, with significantly better performance at manageable computational cost.

## B.5 Motivation of Information Bottleneck for Temporal and Structure Feature Fusion

Temporal graphs require effective fusion of temporal and structural features, which often conflict when combined naively. Our solution adapts the Information Bottleneck method to: (1) create unified node representations, (2) perform data-aware dimensionality reduction, and (3) filter noise while preserving critical temporal-structural information. This approach outperforms naive fusion methods by resolving feature conflicts and optimizing representation quality as shown in 3.

Table 5: Motivation for co-occurrence encoding as structure encoding

|  | CanParl | USLegis | UCI | Untrade |
|---|---|---|---|---|
| MassgePassing | 97.70 | 86.97 | 94.08 | 62.17 |
| RandomWalk | 97.95 | 85.56 | 96.27 | 63.44 |
| Co-occurrence | **99.11** | **92.27** | **96.36** | **93.86** |

Table 6: Trans.AP with different historical neighbor storage.

| | CanParl | USLegis | UCI | Untrade |
|---|---|---|---|---|
| Single Hash | 98.57 | 77.87 | 96.21 | 67.19 |
| Dual Hash | **99.11** | **92.27** | **96.36** | **93.86** |

---

**Algorithm 1:** Structure Memory Update

---

**Data:** Link $(u, v, t)$, hash table size $H$, constants $q, b, p$, sets $S_1^t(u), S_1^t(v)$, memory $M_{struc}^*$
**Result:** Updated $M_{struc}^*$

1 **Function** Hash($a$)**:**
2   |   **return** $((a \cdot q + b) \mod p) \mod H$;
3 $M_{struc}^*[u, \text{Hash}(v)] \leftarrow v$ ;                      // Link $u$ to $v$
4 $M_{struc}^*[v, \text{Hash}(u)] \leftarrow u$ ;                      // Link $v$ to $u$
5 **for** $j \in S_1^t(v)$ **do**
6   |   $M_{struc}^*[u, Hash(j)] \leftarrow j$ ;         // Link $u$ to $v$'s neighbors
7   |   $M_{struc}^*[j, Hash(u)] \leftarrow u$ ;         // Link $v$'s neighbors to $u$
8 **end**
9 **for** $i \in S_1^t(u)$ **do**
10   |   $M_{struc}^*[v, Hash(i)] \leftarrow i$ ;         // Link $v$ to $u$'s neighbors
11   |   $M_{struc}^*[i, Hash(v)] \leftarrow v$ ;         // Link $u$'s neighbors to $v$
12 **end**

---

## C   Complementary Explanation

### C.1   Structure Memory Update

Hash function for a hash table with length $H$ is denoted as $Hash(a) = ((a \cdot q + b) \mod p) \mod H$, where $q$ and $b$ are constants, $p$ is a large enough constant. When a new link $(u, v, t)$ occurs, the structure memory of $u$ and $v$ with each other $M_{struc}^*[u, Hash(v)] \leftarrow v$, $M_{struc}^*[v, Hash(u)] \leftarrow u$ and their neighbors $M_{struc}^*[u, Hash(j)] \leftarrow j, j \in S_1^t(v)$, $M_{struc}^*[v, Hash(i)] \leftarrow i, i \in S_1^t(u)$. And we update the structure memory of neighbor nodes $M_{struc}^*(j, Hash(u)) \leftarrow u, j \in S_1^t(v)$, $M_{struc}^*(i, Hash(v)) \leftarrow v, i \in S_1^t(u)$, where $M_{struc}^l$ and $M_{struc}^s$ indicate two hash tables. We elaborate pseudocode of the structure memory update procedure in Algorithm 1.

For clarity, we further explain why we implement a dual-hash strategy. The dual-memory structure (long-term and short-term) effectively captures both historical and recent neighbor records while reducing hash collision effects. A single hash table would degrade feature extraction, worsen collision impacts, and produce less distinguishable embeddings, ultimately harming performance. To verify, we evaluated Trans.AP of SALoM using a different design for structure memory, including dual-hash and single-hash. Results are shown in Table 6. Using a dual-hash design for structure memory consistently outperforms a single hash design. Verifying the advantage of using a dual-hash structure.

### C.2   Proof of Upper Bound of Information Bottleneck Loss Function

In this section, we provide proof of the upper bound of the information bottleneck loss function. For clarity, we assume feature denoted as $X$, label denoted as $Y$, and the intermediate variable denoted $Z$. We assume the joint distribution $p(X, Y, Z)$ as follows:

$$p(X, Y, Z) = p(Z|X, Y)p(Y|X)p(X) = p(Z|X)p(Y|X)p(X) \tag{20}$$

The IB objective has the form $I(Z, Y) - \beta I(Z, X)$, we examine two terms individually. First for $I(Z, Y)$, writing it in full:

$$I(Z, Y) = \int \text{dydz} \, p(y, z) \log \frac{p(y, z)}{p(y)p(z)} = \int \text{dydz} \, p(y, z) \log \frac{p(y|z)}{p(z)} \tag{21}$$

where $p(y|z)$ is defined by IB Encoder:

$$p(y|z) = \int \text{dx} \, p(x, y|z) = \int \text{dx} \, p(y|x)p(x|z) = \int \text{dx} \, \frac{p(y|x)p(z|x)p(x)}{p(z)} \tag{22}$$

Let $q(y|z)$ be a variational approximation to $p(y|z)$, as the IB Decoder. Using the fact that the Kullback-Leibler divergence is always positive, we have:

$$\text{KL}[p(Y|Z), q(Y|Z)] \geq 0 \Rightarrow \int \mathrm{dy}\, p(y|z)\mathrm{log}p(y|z) \geq \int \mathrm{dy}\, p(y|z)\mathrm{log}q(y|z) \tag{23}$$

and hence

$$
\begin{aligned}
I(Z,Y) &\geq \int \mathrm{dydz}\, p(y,z)\mathrm{log}\frac{q(y|z)}{p(y)} \\
&= \int \mathrm{dydz}\, p(y,z)\mathrm{log}q(y|z) - \int \mathrm{dy}\, p(y)\mathrm{log}p(y) \\
&= \int \mathrm{dydz}\, p(y,z)\mathrm{log}q(y|z) + H(Y)
\end{aligned}
\tag{24}
$$

Note that the entropy of labels $H(Y)$ is independent of optimization objective, so can be ignored. We can rewrite $p(y,z) = \int \mathrm{dx}\, p(x,y,z) = \int \mathrm{dx}\, p(x)p(y|x)p(z|x)$, therefore:

$$I(Z,Y) \geq \int \mathrm{dxdydz}\, p(x)p(y|x)p(z|x)\mathrm{log}q(y|z) \tag{25}$$

Then consider $\beta I(Z,X)$

$$I(Z,X) = \int \mathrm{dzdx}\, p(x,z)\mathrm{log}\frac{p(z|x)}{p(z)} = \int \mathrm{dzdx}\, p(x,z)\mathrm{log}p(z|x) - \int \mathrm{dz}\, p(z)\mathrm{log}p(z) \tag{26}$$

As the marginal distribution of Z is difficult to compute, let r(z) be a variational approximation to the marginal distribution with $\mathcal{N}(0,1)$. Since $\text{KL}[p(Z), r(Z)] \geq 0 \Rightarrow \int \mathrm{dz}\, p(z)\mathrm{log}p(z) \geq \int \mathrm{dz}\, p(z)\mathrm{log}r(z)$, we can summarize the following upper bound:

$$I(Z,X) \leq \int \mathrm{dxdz}\, p(x)p(z|x)\mathrm{log}\frac{p(z|x)}{r(z)} \tag{27}$$

Combining both upper bounds we have:

$$
\begin{aligned}
I(Z,Y) - \beta I(Z,X) &\geq \int \mathrm{dxdydz}\, p(x)p(y|x)p(z|x)\mathrm{log}q(y|z) \\
&\quad - \beta \int \mathrm{dxdz}\, p(x)p(z|x)\mathrm{log}\frac{p(z|x)}{r(z)} = L
\end{aligned}
\tag{28}
$$

In practice, we approximate $p(x,y) = p(y)p(y|x)$ with the empirical data distribution $p(x,y) = \frac{1}{N}\sum_{n=1}^{N}\delta_{x_n}(x)\delta_{y_n}(y)$, and hence we have:

$$L \approx \frac{1}{N}\sum_{n=1}^{N}\left[\int \mathrm{dz}\, p(z|x_n)\mathrm{log}q(y_n|z) - \beta p(z|x_n)\mathrm{log}\frac{p(z|x_n)}{r(z)}\right] \tag{29}$$

The encoder is of form $p(z|x) = \mathcal{N}(z|f_e^\mu(x), f_e^\Sigma(x))$, where $f_e$ is an MLP which outputs both $K$-dimensional mean and the $K \times K$ covariance matrix. Then with reparameterization, $p(z|x)dz = p(\epsilon)d\epsilon$, where $z = f(x,\epsilon)$ is a deterministic function of x and the Gaussian random variable. Assuming our choice of p(z|x) and r(z) allows computation of an analytic Kullback-Leibler divergence, we can have optimization objective:

$$J_{IB} = \frac{1}{N}\sum_{n=1}^{N}\mathbb{E}_{\epsilon \sim p(\epsilon)}\left[-\mathrm{log}q(y_n|f(x_n,\epsilon))\right] + \beta\text{KL}\left[p(Z|x_n), r(Z)\right] \tag{30}$$

### C.3  Leakage-Free TGNNs details

In this section, we give a concrete example. Consider that we maintain two memory units and associated mailboxes, ensuring that the timestamps $t_1$ and $t_2$ of the two memory units satisfy the condition $t_1 < t_2$. In this context, $t_1$ represents the data information that can be observed in real scenarios, while $t_2$ stores the memory unit corresponding to the current prediction time frame. This setup allows us to effectively update $t_1$ for future predictions.

Table 7: Evaluation on leakage-free terms on a subset of datasets.

| Metrics | Datasets | CAWN | TCL | GraphMixer | DyGFormer | SALoM |
|---|---|---|---|---|---|---|
| Trans-AP | MOOC | 80.15 | 82.38 | 82.78 | 87.52 | **91.39** |
| | UCI | 95.18 | 89.57 | 93.25 | 95.79 | **96.39** |
| | Enron | 89.56 | 79.70 | 82.25 | 92.47 | **93.19** |
| | Can. Parl. | 69.82 | 68.67 | 77.04 | 97.36 | **98.79** |
| | US Legis. | 70.58 | 69.59 | 70.74 | 71.11 | **75.79** |
| | UN Trade | 65.39 | 62.21 | 62.61 | 66.46 | **92.52** |
| | UN Vote | 52.84 | 51.90 | 52.11 | 55.55 | **68.52** |
| | Avg. Rank | 3.57 | 4.86 | 3.57 | 2.00 | **1.00** |
| Trans-AUC | MOOC | 80.38 | 83.12 | 84.01 | 87.91 | **91.69** |
| | UCI | 93.87 | 87.82 | 91.81 | 94.49 | **95.46** |
| | Enron | 90.45 | 75.74 | 84.38 | 93.33 | **94.62** |
| | Can. Parl. | 75.70 | 72.46 | 83.17 | 97.76 | **98.75** |
| | US Legis. | 77.16 | 76.27 | 76.96 | 77.90 | **78.84** |
| | UN Trade | 68.54 | 64.72 | 65.52 | 70.20 | **91.59** |
| | UN Vote | 53.09 | 51.88 | 52.46 | 57.12 | **65.60** |
| | Avg. Rank | 3.43 | 4.86 | 3.71 | 2.00 | **1.00** |
| Ind-AP | MOOC | 81.42 | 80.60 | 81.41 | 86.96 | **90.37** |
| | UCI | 92.73 | 87.36 | 91.19 | 94.54 | **94.68** |
| | Enron | 86.35 | 76.14 | 75.88 | 89.76 | **90.42** |
| | Can. Parl. | 55.80 | 54.30 | 55.91 | 87.74 | **95.98** |
| | US Legis. | 53.17 | 52.59 | 50.71 | 54.28 | **62.39** |
| | UN Trade | 65.24 | 62.21 | 62.17 | 64.55 | **82.07** |
| | UN Vote | 49.94 | 51.60 | 50.68 | 55.93 | **70.74** |
| | Avg. Rank | 3.29 | 4.29 | 4.29 | 2.14 | **1.00** |
| Ind-AUC | MOOC | 81.86 | 81.43 | 82.77 | 87.62 | **90.34** |
| | UCI | 90.40 | 84.49 | 89.30 | 92.63 | **92.82** |
| | Enron | 86.35 | 76.14 | 75.88 | 89.76 | **91.61** |
| | Can. Parl. | 58.83 | 55.83 | 58.32 | 89.33 | **95.65** |
| | US Legis. | 51.49 | 50.43 | 47.20 | 53.21 | **55.38** |
| | UN Trade | 67.05 | 63.76 | 63.48 | 67.25 | **79.20** |
| | UN Vote | 48.34 | 50.51 | 50.04 | 56.73 | **67.28** |
| | Avg. Rank | 3.43 | 4.29 | 4.29 | 2.00 | 1.00 |

If for the subsequent event to be predicted $(u, v, t_3)$, we have $t_1 < t_3 = t_2$, then we will use the state recorded at $t_1$ to calculate the memory updater and merge the last event into the memory unit at $t_2$. If $t_1 < t_2 < t_3$, we will use the memory at $t_2$ for model calculations. In this case, the timestamps of subsequent events will be greater than or equal to $t_3$, allowing the memory recorded at $t_2$ to replace that of $t_1$.

This ensures that there is always a memory unit state that is strictly less than the current timestamp and corresponds to the last interaction involved in the calculation. This approach not only avoids the common data leakage issues of traditional TGN but also eliminates the memory and computational overhead of searching for updated states in historical memory records. Additional experimental results validating the effectiveness of this approach are presented in Table 7.

## D    Complementary Experiments

### D.1    Performance Study With Standard Deviates

As for space constraints, the accuracy of model performance with display of standard deviates is shown in Table 8.

We observe that SALoM shows improved stability compared with TGN, which stems from LSMU and Information Bottleneck generating more stable node representations.

### D.2    Training Expense of LSMU

While LSMU does incur higher computational overhead than GRU due to its enhanced capabilities, it maintains reasonable efficiency overall. The GPU memory requirement doubles due to its two-expert selection, but we argue this trade-off is justified by the substantial accuracy improvements achieved.

Table 8: AP&AUC-ROC for transductive and inductive link prediction.

| Metrics | Datasets | JODIE | DyRep | TGAT | TGN | CAWN | EdgeBank | TCL | GraphMixer | NAT | DyGFormer | CNE-N | SALoM |
|---|---|---|---|---|---|---|---|---|---|---|---|---|---|
| Trans\\AP | Wikipedia | 96.50 ± 0.14 | 94.86 ± 0.06 | 96.94 ± 0.06 | 98.28 ± 0.06 | 98.76 ± 0.03 | 90.37 ± 0.00 | 96.47 ± 0.16 | 97.25 ± 0.03 | 97.50 ± 0.04 | **99.03 ± 0.02** | 98.61 ± 0.04 | **99.03 ± 0.02** |
| | Reddit | 98.31 ± 0.14 | 98.22 ± 0.04 | 98.52 ± 0.02 | 98.47 ± 0.06 | 99.11 ± 0.01 | 94.86 ± 0.00 | 97.53 ± 0.02 | 97.31 ± 0.01 | 99.10 ± 0.21 | 99.22 ± 0.01 | 99.26 ± 0.01 | **99.27 ± 0.03** |
| | MOOC | 80.23 ± 2.44 | 81.97 ± 0.49 | 85.84 ± 0.15 | **93.21 ± 1.51** | 80.15 ± 0.25 | 57.97 ± 0.00 | 82.38 ± 0.24 | 82.78 ± 0.15 | 87.21 ± 0.63 | 87.52 ± 0.49 | 90.16 ± 0.07 | 92.42 ± 0.96 |
| | LastFM | 70.85 ± 2.13 | 71.92 ± 2.21 | 73.42 ± 0.21 | 84.36 ± 3.97 | 86.99 ± 0.06 | 79.29 ± 0.00 | 67.27 ± 2.16 | 75.61 ± 0.24 | 88.57 ± 1.76 | 93.00 ± 0.12 | 92.60 ± 0.03 | **93.14 ± 0.35** |
| | Enron | 84.77 ± 0.30 | 82.38 ± 3.36 | 71.12 ± 0.97 | 91.51 ± 1.11 | 89.56 ± 0.09 | 83.53 ± 0.00 | 79.70 ± 0.71 | 82.25 ± 0.16 | 90.81 ± 0.31 | 92.47 ± 0.12 | 92.13 ± 0.06 | **94.08 ± 0.40** |
| | Social Evo. | 89.89 ± 0.55 | 88.87 ± 0.30 | 93.16 ± 0.17 | 89.83 ± 0.17 | 84.96 ± 0.09 | 74.95 ± 0.00 | 93.13 ± 0.16 | 93.37 ± 0.07 | 90.99 ± 0.05 | **94.73 ± 0.01** | 94.50 ± 0.04 | **94.73 ± 0.07** |
| | UCI | 89.43 ± 1.09 | 65.14 ± 2.30 | 79.63 ± 0.70 | 92.94 ± 1.04 | 95.18 ± 0.06 | 76.20 ± 0.00 | 89.57 ± 1.63 | 93.25 ± 0.57 | 94.26 ± 0.37 | 95.79 ± 0.17 | 95.64 ± 0.11 | **96.36 ± 0.05** |
| | Flights | 95.60 ± 1.73 | 95.29 ± 0.72 | 94.03 ± 0.18 | 97.94 ± 0.14 | 98.51 ± 0.01 | 89.35 ± 0.00 | 91.23 ± 0.02 | 90.99 ± 0.05 | 97.66 ± 0.80 | 98.91 ± 0.01 | 98.73 ± 0.01 | **98.94 ± 0.09** |
| | Can. Parl. | 69.26 ± 0.31 | 66.54 ± 2.76 | 70.73 ± 0.72 | 69.29 ± 2.34 | 69.82 ± 2.34 | 64.55 ± 0.00 | 68.67 ± 2.67 | 77.04 ± 0.46 | 83.83 ± 1.20 | 97.36 ± 0.45 | 81.84 ± 2.27 | **99.11 ± 1.52** |
| | US Legis. | 75.05 ± 1.52 | 75.34 ± 0.39 | 68.52 ± 3.16 | 78.09 ± 0.58 | 70.58 ± 0.48 | 58.39 ± 0.00 | 69.59 ± 0.48 | 70.74 ± 1.02 | 77.56 ± 0.21 | 71.11 ± 0.59 | 72.58 ± 0.32 | **92.27 ± 0.63** |
| | UN Trade | 64.94 ± 0.31 | 63.21 ± 0.93 | 61.47 ± 0.18 | 68.30 ± 1.37 | 65.39 ± 0.12 | 60.41 ± 0.00 | 62.21 ± 0.03 | 62.61 ± 0.27 | 72.32 ± 0.69 | 66.46 ± 1.29 | 77.97 ± 0.20 | **93.86 ± 0.55** |
| | UN Vote | 63.91 ± 0.81 | 62.81 ± 0.80 | 52.21 ± 0.98 | 64.13 ± 2.17 | 52.84 ± 0.10 | 58.49 ± 0.00 | 51.90 ± 0.30 | 52.11 ± 0.16 | 69.70 ± 0.49 | 55.55 ± 0.42 | 58.10 ± 0.15 | **86.81 ± 1.43** |
| | Contact | 95.31 ± 1.33 | 95.98 ± 0.15 | 96.28 ± 0.09 | 95.00 ± 0.56 | 90.26 ± 0.28 | 92.58 ± 0.00 | 92.44 ± 0.12 | 91.92 ± 0.03 | 97.25 ± 0.33 | 98.29 ± 0.01 | 98.28 ± 0.01 | **98.53 ± 0.41** |
| | Avg. Rank | 7.76 | 8.61 | 8.38 | 4.92 | 7.07 | 10.53 | 9.76 | 8.38 | 4.53 | 3.15 | 3.61 | **1.076923** |
| Trans\\AUC | Wikipedia | 96.33 ± 0.07 | 94.37 ± 0.09 | 96.67 ± 0.07 | 98.01 ± 0.07 | 98.54 ± 0.04 | 90.78 ± 0.00 | 95.84 ± 0.18 | 96.92 ± 0.03 | 96.72 ± 0.21 | **98.91 ± 0.02** | 98.40 ± 0.06 | 98.87 ± 0.03 |
| | Reddit | 98.31 ± 0.05 | 98.17 ± 0.05 | 98.47 ± 0.02 | 98.32 ± 0.06 | 99.01 ± 0.01 | 95.37 ± 0.00 | 97.42 ± 0.02 | 97.17 ± 0.02 | 99.03 ± 0.02 | 99.15 ± 0.01 | **99.19 ± 0.01** | **99.20 ± 0.03** |
| | MOOC | 83.81 ± 2.09 | 85.03 ± 0.58 | 87.11 ± 0.19 | **93.56 ± 1.01** | 80.38 ± 0.26 | 60.86 ± 0.00 | 83.12 ± 0.18 | 84.01 ± 0.17 | 88.38 ± 0.71 | 87.91 ± 0.58 | 91.42 ± 0.09 | 92.52 ± 0.93 |
| | LastFM | 70.49 ± 1.66 | 71.16 ± 1.89 | 71.59 ± 0.18 | 82.66 ± 2.94 | 85.92 ± 0.10 | 83.77 ± 0.00 | 64.06 ± 1.16 | 73.53 ± 0.12 | 86.94 ± 2.29 | **93.05 ± 0.10** | 92.21 ± 0.03 | 92.32 ± 0.29 |
| | Enron | 87.96 ± 0.52 | 84.89 ± 3.00 | 68.89 ± 1.10 | 90.99 ± 0.99 | 90.45 ± 0.14 | 87.05 ± 0.00 | 75.74 ± 0.72 | 84.38 ± 0.21 | 92.02 ± 0.32 | 93.33 ± 0.13 | 92.77 ± 0.10 | **95.11 ± 0.08** |
| | Social Evo. | 92.05 ± 0.46 | 90.76 ± 0.21 | 94.76 ± 0.16 | 90.36 ± 0.17 | 87.34 ± 0.08 | 81.60 ± 0.00 | 94.84 ± 0.17 | 95.23 ± 0.07 | 93.22 ± 0.13 | **96.30 ± 0.01** | 96.20 ± 0.03 | **96.36 ± 0.08** |
| | UCI | 90.44 ± 0.49 | 68.77 ± 2.34 | 78.53 ± 0.74 | 92.17 ± 1.13 | 93.87 ± 0.08 | 77.30 ± 0.00 | 87.82 ± 1.36 | 91.81 ± 0.67 | 93.02 ± 0.48 | 94.49 ± 0.26 | 94.32 ± 0.16 | **95.53 ± 0.07** |
| | Flights | 96.21 ± 1.42 | 95.95 ± 0.62 | 94.13 ± 0.17 | 97.99 ± 0.13 | 98.45 ± 0.01 | 90.23 ± 0.00 | 91.21 ± 0.02 | 91.13 ± 0.01 | 97.32 ± 0.34 | **98.93 ± 0.01** | 98.74 ± 0.01 | **98.99 ± 0.11** |
| | Can. Parl. | 78.21 ± 0.23 | 73.35 ± 3.67 | 75.69 ± 0.78 | 97.17 ± 1.80 | 75.70 ± 3.27 | 64.14 ± 0.00 | 72.46 ± 3.23 | 83.17 ± 0.53 | 87.70 ± 1.37 | 97.76 ± 0.41 | 84.49 ± 2.47 | **99.18 ± 1.88** |
| | US Legis. | 82.85 ± 1.07 | 82.28 ± 0.32 | 75.84 ± 1.99 | 84.63 ± 0.43 | 77.16 ± 0.39 | 62.57 ± 0.00 | 76.27 ± 0.63 | 76.96 ± 0.79 | 84.68 ± 0.35 | 77.90 ± 0.58 | 79.38 ± 0.29 | **93.75 ± 0.45** |
| | UN Trade | 69.62 ± 0.44 | 67.44 ± 0.83 | 64.01 ± 0.12 | 69.41 ± 1.67 | 68.54 ± 0.18 | 66.75 ± 0.00 | 64.72 ± 0.05 | 65.52 ± 0.51 | 76.76 ± 0.81 | 70.20 ± 1.44 | 79.64 ± 0.14 | **93.23 ± 0.55** |
| | UN Vote | 68.53 ± 0.95 | 67.18 ± 1.04 | 52.83 ± 1.12 | 62.76 ± 2.65 | 53.09 ± 0.22 | 62.97 ± 0.00 | 51.88 ± 0.36 | 52.46 ± 0.27 | 74.44 ± 2.01 | 57.12 ± 0.62 | 60.67 ± 0.14 | **87.87 ± 1.72** |
| | Contact | 96.66 ± 0.89 | 96.48 ± 0.14 | 96.95 ± 0.08 | 95.37 ± 0.35 | 89.99 ± 0.34 | 94.34 ± 0.00 | 94.15 ± 0.09 | 93.94 ± 0.02 | 97.64 ± 0.58 | 98.53 ± 0.01 | 98.62 ± 0.01 | **98.69 ± 0.27** |
| | Avg. Rank | 7.07 | 8.38 | 8.69 | 5.53 | 7.15 | 10.07 | 10.07 | 8.61 | 4.3 | 3.23 | 3.53 | **1.23** |
| Ind\\AP | Wikipedia | 94.82 ± 0.20 | 92.43 ± 0.37 | 96.22 ± 0.07 | 97.49 ± 0.04 | 98.24 ± 0.03 | - | 96.22 ± 0.17 | 96.65 ± 0.02 | 95.40 ± 0.04 | **98.59 ± 0.03** | 97.76 ± 0.06 | 98.49 ± 0.05 |
| | Reddit | 96.50 ± 0.13 | 96.09 ± 0.11 | 97.09 ± 0.04 | 97.26 ± 0.07 | 98.62 ± 0.01 | - | 94.09 ± 0.07 | 95.26 ± 0.02 | 98.56 ± 0.21 | 98.84 ± 0.02 | 98.82 ± 0.03 | **98.93 ± 0.09** |
| | MOOC | 79.63 ± 1.92 | 81.07 ± 0.44 | 85.50 ± 0.19 | **91.86 ± 1.05** | 81.42 ± 0.24 | - | 80.60 ± 0.22 | 81.41 ± 0.21 | 83.59 ± 1.58 | 86.96 ± 0.43 | 88.71 ± 0.04 | 90.53 ± 1.06 |
| | LastFM | 81.61 ± 3.82 | 83.02 ± 1.48 | 78.63 ± 0.31 | 87.18 ± 4.29 | 89.42 ± 0.07 | - | 73.53 ± 1.66 | 82.11 ± 0.42 | 86.87 ± 1.95 | 94.23 ± 0.09 | 94.00 ± 0.05 | **94.56 ± 0.38** |
| | Enron | 80.72 ± 1.39 | 74.55 ± 3.95 | 67.05 ± 1.51 | 84.53 ± 1.02 | 86.35 ± 0.51 | - | 76.14 ± 0.79 | 75.88 ± 0.48 | 89.03 ± 0.83 | 89.76 ± 0.34 | 87.59 ± 0.07 | **91.67 ± 0.08** |
| | Social Evo. | 91.96 ± 0.48 | 90.04 ± 0.47 | 91.41 ± 0.16 | 82.85 ± 0.86 | 79.94 ± 0.18 | - | 91.55 ± 0.09 | 91.86 ± 0.06 | 91.22 ± 0.32 | **93.14 ± 0.04** | 92.70 ± 0.07 | 92.84 ± 0.36 |
| | UCI | 79.86 ± 1.48 | 57.48 ± 1.87 | 79.54 ± 0.48 | 82.04 ± 2.25 | 92.73 ± 0.06 | - | 87.36 ± 2.03 | 91.19 ± 0.42 | 87.30 ± 0.15 | **94.54 ± 0.12** | 93.58 ± 0.03 | 94.36 ± 0.14 |
| | Flights | 94.74 ± 0.37 | 92.88 ± 0.73 | 88.73 ± 0.33 | 95.03 ± 0.60 | 97.06 ± 0.02 | - | 83.41 ± 0.07 | 83.03 ± 0.05 | 96.59 ± 1.67 | 97.79 ± 0.02 | 97.34 ± 0.01 | **97.85 ± 0.12** |
| | Can. Parl. | 53.92 ± 0.94 | 54.02 ± 0.76 | 55.18 ± 0.79 | 78.75 ± 0.93 | 55.80 ± 0.69 | - | 54.30 ± 0.66 | 55.91 ± 0.82 | 60.62 ± 2.26 | 87.74 ± 0.71 | 65.01 ± 1.91 | **96.20 ± 1.25** |
| | US Legis. | 54.93 ± 2.29 | 57.28 ± 0.71 | 51.00 ± 3.11 | 55.74 ± 4.37 | 53.17 ± 1.20 | - | 52.59 ± 0.97 | 50.71 ± 0.76 | 57.54 ± 0.80 | 54.28 ± 2.87 | 59.54 ± 0.33 | **68.38 ± 0.13** |
| | UN Trade | 59.65 ± 0.77 | 57.02 ± 0.69 | 61.03 ± 0.18 | 77.86 ± 3.15 | 65.24 ± 0.21 | - | 62.21 ± 0.12 | 62.17 ± 0.31 | 69.29 ± 1.59 | 64.55 ± 0.62 | 69.84 ± 0.20 | **85.46 ± 0.48** |
| | UN Vote | 56.64 ± 0.96 | 54.62 ± 2.22 | 52.24 ± 1.46 | 65.67 ± 2.51 | 49.94 ± 0.45 | - | 51.60 ± 0.97 | 50.68 ± 0.44 | **66.35 ± 4.06** | 55.93 ± 0.39 | 57.57 ± 0.19 | 62.37 ± 1.74 |
| | Contact | 94.34 ± 1.45 | 92.18 ± 0.41 | 95.87 ± 0.11 | 88.56 ± 0.99 | 89.55 ± 0.30 | - | 91.11 ± 0.12 | 90.59 ± 0.05 | 96.79 ± 0.37 | **98.03 ± 0.02** | 97.58 ± 0.01 | 97.79 ± 0.83 |
| | Avg. Rank | 7.92 | 8.69 | 8.15 | 5.38 | 6.38 | - | 8.53 | 8.15 | 5.07 | 2.84 | 3.23 | **1.53** |
| Ind\\AUC | Wikipedia | 94.33 ± 0.27 | 91.49 ± 0.45 | 95.90 ± 0.09 | 97.08 ± 0.03 | 98.03 ± 0.04 | - | 95.57 ± 0.20 | 96.30 ± 0.04 | 94.74 ± 0.44 | **98.48 ± 0.03** | 97.45 ± 0.11 | 98.26 ± 0.05 |
| | Reddit | 96.52 ± 0.13 | 96.05 ± 0.12 | 96.98 ± 0.04 | 96.94 ± 0.07 | 98.42 ± 0.02 | - | 93.80 ± 0.07 | 94.97 ± 0.05 | 97.99 ± 0.52 | 98.71 ± 0.01 | 98.69 ± 0.03 | **98.85 ± 0.09** |
| | MOOC | 83.16 ± 1.30 | 84.03 ± 0.49 | 86.84 ± 0.17 | **92.02 ± 0.91** | 81.86 ± 0.25 | - | 81.43 ± 0.19 | 82.77 ± 0.24 | 86.13 ± 3.55 | 87.62 ± 0.51 | 89.94 ± 0.04 | 90.09 ± 0.99 |
| | LastFM | 81.13 ± 3.39 | 82.24 ± 1.51 | 76.99 ± 0.29 | 85.58 ± 3.15 | 87.82 ± 0.12 | - | 70.84 ± 0.85 | 80.37 ± 0.18 | 83.07 ± 2.32 | **94.08 ± 0.08** | 93.62 ± 0.04 | 93.77 ± 0.38 |
| | Enron | 81.96 ± 1.34 | 76.34 ± 4.20 | 64.63 ± 1.74 | 83.58 ± 1.11 | 87.02 ± 0.50 | - | 72.33 ± 0.99 | 76.51 ± 0.71 | 89.92 ± 0.72 | 90.69 ± 0.26 | 88.24 ± 0.07 | **92.57 ± 0.18** |
| | Social Evo. | 93.70 ± 0.29 | 91.18 ± 0.49 | 93.41 ± 0.19 | 82.04 ± 0.59 | 84.73 ± 0.27 | - | 93.71 ± 0.18 | 94.09 ± 0.07 | 92.11 ± 0.07 | **95.29 ± 0.003** | 94.99 ± 0.03 | 95.03 ± 0.17 |
| | UCI | 78.80 ± 0.94 | 58.08 ± 1.81 | 77.64 ± 0.38 | 86.48 ± 2.29 | 90.40 ± 0.11 | - | 84.49 ± 1.82 | 89.30 ± 0.57 | 83.81 ± 1.28 | **92.63 ± 0.13** | 91.31 ± 0.11 | 92.17 ± 0.17 |
| | Flights | 95.21 ± 0.32 | 93.56 ± 0.70 | 88.64 ± 0.35 | 95.92 ± 0.43 | 96.86 ± 0.02 | - | 82.48 ± 0.01 | 82.27 ± 0.06 | 96.36 ± 1.51 | 97.80 ± 0.02 | 97.20 ± 0.01 | **97.95 ± 0.27** |
| | Can. Parl. | 53.81 ± 1.14 | 55.27 ± 0.49 | 56.51 ± 0.75 | 80.21 ± 0.75 | 58.83 ± 1.13 | - | 55.83 ± 1.07 | 58.32 ± 1.08 | 61.62 ± 2.50 | 89.33 ± 0.48 | 66.51 ± 1.25 | **96.07 ± 1.65** |
| | US Legis. | 58.12 ± 2.35 | 61.07 ± 0.56 | 48.27 ± 3.50 | 58.87 ± 4.48 | 51.49 ± 1.13 | - | 50.43 ± 1.48 | 47.20 ± 0.89 | 62.85 ± 0.84 | 53.21 ± 3.04 | 60.10 ± 0.43 | **65.56 ± 0.15** |
| | UN Trade | 62.28 ± 0.50 | 58.82 ± 0.98 | 62.72 ± 0.12 | 75.70 ± 3.50 | 67.05 ± 0.21 | - | 63.76 ± 0.07 | 63.48 ± 0.37 | 72.56 ± 1.47 | 67.25 ± 1.05 | 71.40 ± 0.20 | **83.04 ± 0.23** |
| | UN Vote | 58.13 ± 1.43 | 55.13 ± 3.46 | 51.83 ± 1.35 | 61.64 ± 2.71 | 48.34 ± 0.76 | - | 50.51 ± 1.05 | 50.04 ± 0.86 | **66.26 ± 5.48** | 56.73 ± 0.69 | 58.85 ± 0.24 | 62.44 ± 1.92 |
| | Contact | 95.37 ± 0.92 | 91.89 ± 0.38 | 96.53 ± 0.10 | 88.87 ± 0.75 | 89.07 ± 0.34 | - | 93.05 ± 0.09 | 92.83 ± 0.05 | 96.67 ± 0.45 | 98.30 ± 0.02 | 97.91 ± 0.01 | **97.98 ± 0.67** |
| | Avg. Rank | 7.76 | 8.53 | 8.07 | 5.46 | 6.53 | - | 8.76 | 8.15 | 5.00 | 2.69 | 3.46 | **1.53** |

Table 9: Evaluation of the implantation of LSMU on TGN.

| | | **LSMU** | **GRU** |
|---|---|---|---|
| CanParl | routing | 31.95s | - |
| | update | 23.54s | 20.49s |
| | combine | 1.01s | - |
| | total | 56.5s | 20.49s |
| USLegis | routing | 24.55s | - |
| | update | 12.13s | 13.95s |
| | combine | 0.81s | - |
| | total | 37.49s | 13.95s |
| UCI | routing | 24.64s | - |
| | update | 19.08s | 20.09s |
| | combine | 0.81s | - |
| | total | 44.53s | 20.09s |
| Untrade | routing | 198.09s | - |
| | update | 136.79s | 161.78s |
| | combine | 6.60s | - |
| | total | 341.48s | 161.78s |
| GPU Memory | | 18.67M | 9.25M |

Table 10: Evaluation of the implantation of LSMU on TGN.

| | CanParl | | | | USLegis | | | | Untrade | | | |
|---|---|---|---|---|---|---|---|---|---|---|---|---|
| | Trans.AP | Trans.AUC | Ind.AP | Ind.AUC | Trans.AP | Trans.AUC | Ind.AP | Ind.AUC | Trans.AP | Trans.AUC | Ind.AP | Ind.AUC |
| GRU | 61.13 | 69.61 | 52.7 | 53.98 | 74.89 | 82.38 | 61.87 | 64.47 | 60.15 | 63.69 | 58.34 | 58.11 |
| LSMU | **66.16** | **74.52** | **53.51** | **54.22** | **75.85** | **83.69** | **62.07** | **65.03** | **65.4** | **68.93** | **58.42** | **60.54** |

Table 11: The impact of batch size on model performance.

| | CanParl | | USLegis | | Untrade | |
|---|---|---|---|---|---|---|
| | TGN | SALoM | TGN | SALoM | TGN | SALoM |
| bs=10 | **98.25** | **99.11** | **89.8** | **93.75** | **69.41** | **93.86** |
| bs=20 | 80.95 | 97.79 | 85.32 | 87.99 | 65.24 | 90.57 |
| bs=100 | 68.02 | 89.76 | 76.08 | 77.1 | 64.21 | 79.5 |
| bs=200 | 61.13 | 80.13 | 74.89 | 75.91 | 60.15 | 72.84 |
| bs=1000 | 61.87 | 72.76 | 70.4 | 71.36 | 62.88 | 65.57 |

### D.3 LSMU as a Universal Memory Updater

Since LSMU is designed for the memory update process, it serves as a universal memory updater for all memory-based methods. Notably, LSMU was initially developed on TGN; therefore can allow straightforward adaptation to other memory-based methods for accuracy improvements. We also extended our experiments on TGN with the default setting and only alteration of the memory updater to support this view as follows.

### D.4 The Impact of Batch Size on Model Performance

We evaluated the transductive AP of SALoM and TGN with different batch sizes to verify the impact of batch size on model performance.

As shown in Table 11, model performance degrades as batch size increases. Predict accuracy peaks at batch size 10.

This stems from the inner-batch information loss issue shared by memory-based methods. While training optimizations could mitigate this, our focus remains on model design.

### D.5 Configurations of SALoM Hyper-Parameters

- Number of sampled neighbors: 10
- Dimension of encoder time interval: 100
- Dimension of node temporal memory: 172
- Dimension of node structure memory: 64
- Dimension of output representation: 172
- Number of experts in LSMU: 6
- Number of selected experts in LSMU: 2
- $\beta$ of information bottleneck loss: $1e^{-3}$

## E Broader Impacts

Our research on optimizing continuous-time dynamic graph models for edge prediction has significant potential to impact various domains where temporal network dynamics are critical. By improving the accuracy and efficiency of predicting evolving connections in graphs, this work can enhance applications in social network analysis, recommendation systems, epidemiological modeling, and financial network forecasting. For instance, better edge prediction could enable more precise identification of emerging social trends, improve personalized content delivery, or enhance the tracking of disease spread in real time. In financial systems, it could aid in detecting fraudulent transactions by modeling dynamic interaction patterns.

By advancing the state-of-the-art in dynamic graph modeling, our research contributes to a deeper understanding of temporal network structures, fostering innovations that are both socially beneficial and ethically responsible.

