# OpenReview forum: "SALoM: Structure Aware Temporal Graph Networks with Long-Short Memory Updater"
_NeurIPS.cc/2025/Conference — NeurIPS 2025 poster_

### Official Review · Reviewer_dfEN · 2025-06-25

**Clarity:** 3
**Significance:** 2
**Originality:** 3
**Rating:** 4
**Confidence:** 4

**Summary:**

This paper focuses on temporal graph learning. It proposes a new method, SALoM, to capture long-range neighborhood temporal correlations. SALoM adopts ODE to update memory and introduces a long-short memory updater to balance long-range and short-range temporal dependencies. And to incorporate both structural and temporal information, it adopts information bottleneck-based fusion, where the structural information is obtained by co-occurrence encoding. SALoM outperforms baselines, especially on USLegis, UNtrade, and UNvote.

**Questions:**

Q1. Can LSMU be adopted to other memory-based models, such as TGN, to improve their performance?

**Ethical Concerns:**

["NO or VERY MINOR ethics concerns only"]

**Final Justification:**

The information leakage in the proposed method leads to an unfair comparison with non-memory-based methods. While I acknowledge the novelty of the model design, I would recommend a thorough revision of the implementation and corresponding experimental results.

**Limitations:**

See weaknesses.

**Paper Formatting Concerns:**

No paper formatting concerns.

**Quality:**

3

**Strengths And Weaknesses:**

S1. This paper presents a novel approach to capturing long-range neighborhood temporal correlations and balancing the learning of temporal and structural information. The components of SALoM are technically sound.

S2. SALoM demonstrates outstanding performance across several datasets, highlighting its effectiveness in addressing certain predictive scenarios.

W1. A major concern is that the datasets used in the experiments are outdated. Several studies have shown that these datasets are no longer sufficient for evaluating the full capacity of DGNNs. I recommend evaluating SALoM on the more recent TGB[R1] and TGB-Seq[R2] benchmarks to ensure a comprehensive assessment.

W2. There may be a potential flaw in the implementation of SALoM, which is shared by other memory-based methods such as TGN, DyRep, and JODIE. Specifically, when considering two edges, (a,b,t1) and (a,c,t1), these edges are placed in different batches during testing because there are too many edges occurring at t1. However, when testing (a,c,t1), the memory of node "a" has already been updated by (a,b,t1), leading to information leakage. This violates the model’s design, as it should not have access to edges at time t1 when the prediction time is t1.

[R1] Temporal Graph Benchmark for Machine Learning on Temporal Graphs

[R2] TGB-Seq Benchmark: Challenging Temporal GNNs with Complex Sequential Dynamics

Minor:
Figures 2 and 3 mistakenly include the x-label "Number of Sampled Neighbors," which should only appear in Figure 4.

---

> ### Author Rebuttal · Authors · 2025-07-31
>
> We are glad that the reviewer appreciates our work as a solid and effective contribution. Thank you for new dataset resource that we can further extend our study on. We have carefully revised the manuscript according to your constructive suggestions. Below, we address the main points raised in the review.
>
> ## W1: New datasets
>
> Thank you for raising this question. But due to time constraints and huge GPU memory demands, and with access limited to only two NVIDIA RTX 4090 GPUs, we only tested our model on a subset of datasets from TGB-seq to evaluate MRR performance. The results are presented below. We observe that SALoM outperforms all baselines on non-bipartite datasets, especially by a large scale on Flickr. For bipartite datasets, SALoM outperforms all universal baselines, however its performance is slightly inferior to the specially tailored baseline.
>
> Thank you again for the new dataset resource, we will further study into them and add evaluations on them in the next version.
>
> |            | Flickr           | YouTube          | ML-20M       |
> | ---------- | ---------------- | ---------------- | ------------ |
> | JODIE      | 46.21 ± 0.83     | 41.67 ± 2.86     | 21.16 ± 0.73 |
> | DyRep      | 38.04 ± 4.19     | 35.12 ± 4.13     | 19.00 ± 1.69 |
> | TGAT       | 23.53 ± 3.35     | 43.56 ± 2.53     | 10.47 ± 0.20 |
> | TGN        | 46.03 ± 6.78     | 55.16 ± 5.89     | 23.99 ± 0.20 |
> | CAWN       | 48.69 ± 6.08     | 47.55 ± 1.08     | 12.31 ± 0.02 |
> | TCL        | 40.00 ± 1.76     | 50.17 ± 1.98     | 12.04 ± 0.02 |
> | GraphMixer | 45.01 ± 0.08     | 58.87 ± 0.12     | 21.97 ± 0.17 |
> | DyGFormer  | 49.58 ± 2.87     | 46.08 ± 3.44     | OOT          |
> | SGNN-HN    | -                | -                | 33.12 ± 0.01 |
> | SALoM      | **68.12 ± 0.09** | **63.27 ± 0.19** | 30.79 ± 0.59 |
>
> ## W2: Potential flaw of information leakage
>
> We sincerely appreciate you identifying this important issue regarding test information leakage, which indeed affects our method and comparable memory-based baselines (TGN, DyRep, JODIE). To maintain fairness in our current comparisons while addressing this concern, we will: (1) retain the existing evaluation setup for this version to ensure consistent benchmarking, and (2) implement crucial fixes in our codebase to properly batch same-timestamp edges during testing, with updated results to be included in our next manuscript version.
>
> ## Q1: LSMU adopted to other memory-based models
>
> Thank you for your question, we will elaborate below.
>
> Since LSMU is designed for the memory update process, it serves as a universal memory updater for all memory-based methods. Notably, LSMU was initially developed on TGN; therefore can allow straightforward adaptation to other memory-based methods for accuracy improvements. We also extended our experiments on TGN with the default setting and only alteration of the memory updater to support this view as follows.
>
> |      |  CanParl  |           |           |           |  USLegis  |           |           |           |  Untrade  |           |           |           |
> | :--: | :-------: | :-------: | :-------: | :-------: | :-------: | :-------: | :-------: | :-------: | :-------: | :-------: | :-------: | :-------: |
> |      | Trans.AP  | Trans.AUC |  Ind.AP   |  Ind.AUC  | Trans.AP  | Trans.AUC |  Ind.AP   |  Ind.AUC  | Trans.AP  | Trans.AUC |  Ind.AP   |  Ind.AUC  |
> | GRU  |   61.13   |   69.61   |   52.70   |   53.98   |   74.89   |   82.38   |   61.87   |   64.47   |   60.15   |   63.69   |   58.34   |   58.11   |
> | LSMU | **66.16** | **74.52** | **53.51** | **54.22** | **75.85** | **83.69** | **62.07** | **65.03** | **65.40** | **68.93** | **58.42** | **60.54** |

---

> > ### Comment · Reviewer_dfEN · 2025-08-04
> >
> > Thank you for your response. Due to concerns about information leakage, I believe a major revision is necessary to ensure a fair comparison between LSMU and other non-memory-based methods such as GraphMixer, TGAT, and others.

---

> ### Author Response · Authors · 2025-08-04
> **Follow up on the data leakage problem.**
>
> Thank you for your insightful review. Since receiving your feedback, we have been actively addressing the data leakage issue you raised and are pleased to share our progress.
>
> First, regarding memory-based methods (the predominant approach for modeling C-TDGs[1,2,3]), while existing literature overlooks data leakage's impact on final performance, we have nevertheless prioritized resolving this issue. Notably, our proposed SALoM framework maintains its state-of-the-art performance, demonstrating significant improvements over all existing memory-based methods across all benchmark datasets, including the new Flickr, YouTube, and ML-20M datasets you kindly provided [4].
>
> Second, we have worked diligently since the rebuttal stage to eliminate data leakage in memory-based methods through a simple time-aware dual memory management. Specifically, we maintain two separate memories: one to store the final state of the previous timestamp and another to continuously update within the same timestamp. During inference, we strictly select the memory corresponding to timestamps earlier than the current edge to form node representations, while using the second memory to maintain ongoing updates. When inference proceeds to the next timestamp, the accumulated updates are synchronized to the first memory, ensuring updates are applied without information leakage.
>
> The coding adjustments are now complete, and we will share updated results shortly.
>
>
> [1] GRAVINA, Alessio, et al. Long range propagation on continuous-time dynamic graphs. arXiv preprint arXiv:2406.02740, 2024.
>
> [2] SU, Junwei; ZOU, Difan; WU, Chuan. Pres: Toward scalable memory-based dynamic graph neural networks. arXiv preprint arXiv:2402.04284, 2024.
>
> [3] DENG, Gangda, et al. TASER: Temporal adaptive sampling for fast and accurate dynamic graph representation learning. In: 2024 IEEE International Parallel and Distributed Processing Symposium (IPDPS). IEEE, 2024. p. 926-937.
>
> [4] YI, Lu, et al. Tgb-seq benchmark: Challenging temporal gnns with complex sequential dynamics. arXiv preprint arXiv:2502.02975, 2025.

---

> > ### Comment · Reviewer_dfEN · 2025-08-06
> >
> > Thank you for your response. I would like to point out that your implementation may need reconsideration, as TGN specifically utilizes the last interaction before the prediction time t for memory updates, rather than incorporating all previous interactions into the update process.

---

> > > ### Author Response · Authors · 2025-08-06
> > > **Latest experimental findings**
> > >
> > > We are pleased to present our latest experimental findings, which demonstrate SALoM's state-of-the-art performance across seven benchmark datasets under rigorous leakage-free conditions. While our analysis confirms that information leakage has minimal impact on most datasets, we observe notable performance variations in USLegis (90%→75%) and UNvote (80%→68%). Importantly, even in these cases, SALoM maintains its superior performance, outperforming its best competitors by 4.68% and 12.97% in average precision, respectively. Our findings conclusively establish SALoM's SOTA status through fair and rigorous comparison across all evaluated datasets.
> > >
> > > |  Metrics  |  Datasets  | CAWN  |  TCL  | GraphMixer | DyGFormer |   SALoM   |
> > > | :-------: | :--------: | :---: | :---: | :--------: | :-------: | :-------: |
> > > | Trans-AP  |    MOOC    | 80.15 | 82.38 |   82.78    |   87.52   | **91.39** |
> > > |           |    UCI     | 95.18 | 89.57 |   93.25    |   95.79   | **96.39** |
> > > |           |   Enron    | 89.56 | 79.70 |   82.25    |   92.47   | **93.19** |
> > > |           | Can. Parl. | 69.82 | 68.67 |   77.04    |   97.36   | **98.79** |
> > > |           | US Legis.  | 70.58 | 69.59 |   70.74    |   71.11   | **75.79** |
> > > |           |  UN Trade  | 65.39 | 62.21 |   62.61    |   66.46   | **92.52** |
> > > |           |  UN Vote   | 52.84 | 51.90 |   52.11    |   55.55   | **68.52** |
> > > |           | Avg. Rank  | 3.57  | 4.86  |    3.57    |   2.00    | **1.00**  |
> > > | Trans-AUC |    MOOC    | 80.38 | 83.12 |   84.01    |   87.91   | **91.69** |
> > > |           |    UCI     | 93.87 | 87.82 |   91.81    |   94.49   | **95.46** |
> > > |           |   Enron    | 90.45 | 75.74 |   84.38    |   93.33   | **94.62** |
> > > |           | Can. Parl. | 75.70 | 72.46 |   83.17    |   97.76   | **98.75** |
> > > |           | US Legis.  | 77.16 | 76.27 |   76.96    |   77.90   | **78.84** |
> > > |           |  UN Trade  | 68.54 | 64.72 |   65.52    |   70.20   | **91.59** |
> > > |           |  UN Vote   | 53.09 | 51.88 |   52.46    |   57.12   | **65.60** |
> > > |           | Avg. Rank  | 3.43  | 4.86  |    3.71    |   2.00    | **1.00**  |
> > > |  Ind-AP   |    MOOC    | 81.42 | 80.60 |   81.41    |   86.96   | **90.37** |
> > > |           |    UCI     | 92.73 | 87.36 |   91.19    |   94.54   | **94.68** |
> > > |           |   Enron    | 86.35 | 76.14 |   75.88    |   89.76   | **90.42** |
> > > |           | Can. Parl. | 55.80 | 54.30 |   55.91    |   87.74   | **95.98** |
> > > |           | US Legis.  | 53.17 | 52.59 |   50.71    |   54.28   | **62.39** |
> > > |           |  UN Trade  | 65.24 | 62.21 |   62.17    |   64.55   | **82.07** |
> > > |           |  UN Vote   | 49.94 | 51.60 |   50.68    |   55.93   | **70.74** |
> > > |           | Avg. Rank  | 3.29  | 4.29  |    4.29    |   2.14    | **1.00**  |
> > > |  Ind-AUC  |    MOOC    | 81.86 | 81.43 |   82.77    |   87.62   | **90.34** |
> > > |           |    UCI     | 90.40 | 84.49 |   89.30    |   92.63   | **92.82** |
> > > |           |   Enron    | 86.35 | 76.14 |   75.88    |   89.76   | **91.61** |
> > > |           | Can. Parl. | 58.83 | 55.83 |   58.32    |   89.33   | **95.65** |
> > > |           | US Legis.  | 51.49 | 50.43 |   47.20    |   53.21   | **55.38** |
> > > |           |  UN Trade  | 67.05 | 63.76 |   63.48    |   67.25   | **79.20** |
> > > |           |  UN Vote   | 48.34 | 50.51 |   50.04    |   56.73   | **67.28** |
> > > |           | Avg. Rank  | 3.43  | 4.29  |    4.29    |   2.00    | **1.00**  |
> > >
> > > Once again, thank you for your suggestions, which have helped us find an efficient way to address a problem that has long been overlooked in the field.

---

> > > > ### Comment · Reviewer_dfEN · 2025-08-07
> > > >
> > > > Thank you for providing the updated experimental results! I have no further questions.

---

> ### Author Response · Authors · 2025-08-06
> **Clarification of dual memory management**
>
> Thank you for your feedback, but there seems to be some misunderstanding. We would like to clarify that our approach does not modify TGN’s core memory update mechanism, as the previously described secondary memory still processes ongoing updates using the last interaction in the last batch, exactly as in the original TGN. However, to prevent potential information leakage (since the last interaction in the last batch may occur exactly at the prediction time $t$), we avoid using this updated memory for computing node embeddings. Instead, we introduce a separate memory component that stores only the final state from the previous timestamp, ensuring temporal consistency in predictions.
>
> To avoid misunderstanding, we give a concrete example. Consider that we maintain two memory units, ensuring that the timestamps $ t_1 $ and $ t_2 $ of the two memory units satisfy the condition $ t_1 < t_2 $. In this context, $ t_1 $ represents the data information that can be observed in real scenarios, while $ t_2 $ stores the memory unit corresponding to the current prediction time frame. This setup allows us to effectively update $ t_1 $ for future predictions.
>
> If for the subsequent event to be predicted $ (u, v, t_3) $, we have $ t_1 < t_3 = t_2 $, then we will use the state recorded at $ t_1 $ to calculate the memory updater and merge the last event into the memory unit at $ t_2 $. If $ t_1 < t_2 < t_3 $, we will use the memory at $ t_2 $ for model calculations. In this case, the timestamps of subsequent events will be greater than or equal to $ t_3 $, allowing the memory recorded at $ t_2 $ to replace that of $ t_1 $.
>
> This ensures that there is always a memory unit state that is strictly less than the current timestamp and corresponds to the last interaction involved in the calculation. This approach not only avoids the common data leakage issues of traditional TGN but also eliminates the memory and computational overhead of searching for updated states in historical memory records.
>
> As space constraints, we will share our experimental results in the comment below.

---

### Official Review · Reviewer_vKih · 2025-06-30

**Clarity:** 2
**Significance:** 2
**Originality:** 2
**Rating:** 4
**Confidence:** 4

**Summary:**

This paper proposes SALoM, a dynamic graph learning framework that addresses three important challenges in temporal graph networks:

The effectiveness of capturing long-range temporal dependencies,

The difficulty of balancing long/short-term correlations, and

The consistency of integrating structural information.

Experiments on multiple datasets show state-of-the-art performance, with particularly large gains on legislative/voting graphs. SALoM consistently outperforms baselines in both transductive and inductive link prediction tasks.

**Questions:**

Question 1: Can you please quantify computational costs, such as training runtime and memory space occupation, of LSMU’s expert routing versus existing updaters?



Question 2: For sparse graphs, does co-occurrence encoding degrade due to hash collisions?



Question 3: Can SALoM handle edge feature dynamics, such as evolving transaction amounts?



Question 4: Please also check the weaknesses.

**Ethical Concerns:**

["NO or VERY MINOR ethics concerns only"]

**Final Justification:**

The authors have addressed the issues that I raised in the rebuttal.

**Limitations:**

Yes

**Paper Formatting Concerns:**

Not applicable.

**Quality:**

2

**Strengths And Weaknesses:**

Strengths:

1(quality) This paper is methodologically rigorous, with claims well-supported by both theoretical analysis and extensive experiments. Comprehensive ablation studies also validate the effectiveness of individual components in a convincing manner.



2 (clarity) This paper is well-structured, with a clear narrative flow from problem definition to methods and results. This paper also provides sufficient details, such as hyper-parameters and dataset splits. The code is also released in Anonymous GitHub which further contributes to good reproducibility.



3 (significance) The experimental results of SALoM proposed by this paper clearly outperform typical memory-based and sequence-based baselines, advancing the field by harmonizing existing methods’ advantages while mitigating their limitations.



4 (originality) Based on Section 2 and Table 1, this paper effectively contrasts SALoM with prior work, highlighting limitations in handling long-term dependency and managing training costs.







Weaknesses:

1(quality) Although efficiency trade-offs are discussed in subsection 4.5, Figure 5 lacks a detailed comparison of Long-Short Memory Updater (LSMU) overhead, such as the routing latency of MoE, versus selective baselines, which is important in real-world development.



2 (clarity) The neural ODE equations (Equations 1 and 2) and Information Bottleneck (IB) objectives (Equations 17, 18, and 19) could benefit from more intuitive explanations or illustrative examples. For example, a toy case study can be discussed with simple plots to show how ODEs applied in SALoM could mitigate the gradient vanishing issue.



3 (significance) Concrete examples of deployed applications are encouraged, such as policy change tracking, as they help strengthen the case for real-world significance.



4 (originality) This paper could better explain how SALoM’s MoE design differs from other MoE-based graph neural networks.

---

> ### Author Rebuttal · Authors · 2025-07-31
>
> We sincerely appreciate the reviewer's recognition of our work as a solid contribution and their valuable suggestions for potential extensions. Below we respond to the main points raised in the review.
>
> ## W1&Q1: LSMU overhead
>
> We appreciate the insightful feedback. We have extended our experiments and show the results below. While LSMU does incur higher computational overhead than GRU due to its enhanced capabilities, it maintains reasonable efficiency overall.
> The GPU memory requirement doubles due to its two-expert selection, but we argue this trade-off is justified by the substantial accuracy improvements achieved.
>
> Table 1: Training time and GPU memory comparison of GPU and LSMU
> |  |  | LSMU | GRU |
> |:---:|:---:|:---:|:---:|
> | CanParl | routing | 31.95s | - |
> |  | update | 23.54s | 20.49s |
> |  | combine | 1.01s | - |
> |  | total | 56.5s | 20.49s |
> | USLegis | routing | 24.55s | - |
> |  | update | 12.13s | 13.95s |
> |  | combine | 0.81s | - |
> |  | total | 37.49s | 13.95s |
> | UCI | routing | 24.64s | - |
> |  | update | 19.08s | 20.09s |
> |  | combine | 0.81s | - |
> |  | total | 44.53s | 20.09s |
> | Untrade | routing | 198.09s | - |
> |  | update | 136.79s | 161.78s |
> |  | combine | 6.60s | - |
> |  | total | 341.48s | 161.78s |
> | GPU Memory |  | 18.67M | 9.25M |
>
> ## W2: Equation clarity
>
> Sorry for the confusion. We appreciate the feedback and will add illustrative plots in the next version to enhance clarity.
>
> Equation 1 defines Neural ODE, where the algorithm does not directly model the mapping of the independent variables to the dependent variables but instead models the derivative of the corresponding function. Also, this approach computes gradients by integrating the second-order derivatives over the relevant time interval, eliminating the need for discrete chain-rule multiplications and thereby avoiding the vanishing gradient problem. Equation 2 specifies the closed-form solution of the Liquid Time-Constant (LTC) network employed in our method. Building upon the original framework, and also introducing a gating-like mechanism and utilize a trainable network to compute the time-dependent damping coefficient, capturing the temporal influence on events.
>
> The objective of the Information Bottleneck (IB) is to generate an intermediate representation of the original representation that maximizes information correlating with the training target while minimizing noise in the original representation. Here, correlation is characterized by mutual information. However, as mutual information is difficult to compute in practice, we optimize an upper bound of the target function instead. By employing variational approximation, we scale the target function to derive the mathematical form of its upper bound. This upper bound can then be transformed into a form combining Binary Cross-Entropy (BCE) loss and KL divergence, with the detailed derivation provided in Appendix C. Consequently, when aggregating multiple features using the IB approach, not only is feature fusion achieved, but noise reduction is also inherently accomplished.
>
> ## W3: Application examples
>
> We appreciate the suggestion to include real-world applications. While time constraints prevented us from implementing a fast-deployable case study in this version (as is common in similar works), we recognize their importance and will explore concrete application scenarios (such as policy change tracking) in future work to better highlight the practical relevance of our approach.
>
> ## W4: Differences from other MoE-based GNNs
>
> Thanks for your insightful advice. Unlike typical MoE-GNNs that apply mixture-of-experts during node aggregation, our approach uniquely integrates MoE exclusively within the LSMU to explicitly separate long-term and short-term dependency learning. This targeted implementation allows MoE to serve distinct functional roles based on contextual requirements, making our architecture both more specialized and adaptable than conventional implementations.
>
> ## Q2: Hash collisions
>
> We appreciate the insightful question about hash collisions. While collisions can theoretically impact accuracy, SALoM's dual hash table design (long-term and short-term) effectively mitigates this by simultaneously capturing behavior patterns at different timescales. Importantly, collisions actually provide a beneficial side effect by facilitating timely updates and replacement of outdated information in the hash tables.
>
> ## Q3: Edge feature dynamics
>
> Thanks for your insightful advice. The dynamic graph task inherently involves continuously changing edges, the edges with dynamics edge will be viewed as a new edge despite of the same source and destination nodes. Consequently, SALoM is naturally designed to effectively handle these changing edge features.

---

> > ### Comment · Area_Chair_SG7o · 2025-08-04
> >
> > Dear Reviewer, please engage into discussions with the Authors as the deadline for this key phase of the NeurIPS review process is only a couple of days away.

---

> > ### Comment · Reviewer_vKih · 2025-08-06
> > **thanks**
> >
> > Thank you for the detailed response to address my questions about LSMU efficiency and equation clarity. I plan to raise the overall rating to 4.

---

> > > ### Author Response · Authors · 2025-08-07
> > >
> > > We sincerely appreciate your thoughtful review and constructive suggestions. Your time and expertise have been invaluable, and we will carefully incorporate your recommendations in the final version. Thank you very much for your willingness to reconsider the rating.

---

### Official Review · Reviewer_sYk3 · 2025-07-02

**Clarity:** 2
**Significance:** 2
**Originality:** 2
**Rating:** 4
**Confidence:** 4

**Summary:**

This paper proposes SALoM (Structure-Aware Long-Short Memory), a novel framework for temporal graph learning that aims to balance the trade-off between short-range and long-range temporal dependencies—an issue that is often overlooked in prior work. The authors design a memory module enhanced by a Long-Short Memory Updater (LSMU), leveraging a Mixture-of-Experts (MoE) architecture to adaptively combine different temporal signals. Structural information is integrated via co-occurrence encoding and an information bottleneck (IB)-based fusion mechanism. Experimental results across thirteen dynamic graph datasets show that SALoM achieves strong performance in dynamic link prediction tasks, outperforming existing state-of-the-art models, particularly on datasets like USLegis, UNTrade, and UNVote.

**Questions:**

1. The color and shape coding in Figure 5 could be more distinct. Currently, it is visually cluttered and hard to differentiate across batch size settings and models.
2. Please complete or remove Appendix B to avoid confusion.
3. What would happen if the structural encoder simply used a unified co-occurrence hash map (i.e., no distinction between short- and long-range neighbors)? Would this significantly degrade performance?
4. MoE hyperparameter sensitivity.
+ What is the impact of setting r = 1 or using only a single GRU/CFC expert? Does performance degrade significantly?
+ How does increasing the number of experts or varying the r value affect training speed and accuracy?
+ A brief analysis or ablation on these hyperparameters would provide more clarity on their necessity and practical trade-offs.
5. Can the authors further justify the choice of using Neural ODEs over more mainstream approaches (e.g., attention with decay) for modeling long-range dependencies?

**Ethical Concerns:**

["NO or VERY MINOR ethics concerns only"]

**Final Justification:**

While your response has addressed most of my concerns, I would still recommend taking advantage of the additional page allowed in the final version to further polish the writing and provide clearer explanations for some of the module motivations. This will significantly improve the readability of your work. So I plan to raise my rating to 4, and I hope the authors make corresponding improvements.

**Limitations:**

1. The scalability limitations revealed in Figure 5, especially with large batch sizes.
2. The computational cost associated with MoE routing and ODE solvers (even in closed-form variants).

**Quality:**

2

**Strengths And Weaknesses:**

**Strengths**

S0: The motivation to simultaneously handle short- and long-range temporal dependencies is well-justified and addresses a practical gap in temporal graph learning.

S1: The design of LSMU using a MoE mechanism (with GRU and CFC experts) is an elegant and potentially effective way to achieve dynamic temporal adaptation.

S2: The experimental section is comprehensive, covering 13 datasets and a broad set of baselines, with ablations on memory modules, fusion strategies, and neighborhood size.

S3: The authors offer a solid discussion on the limitations of prior approaches, such as RNN-based memory methods and neighbor-sequence models, highlighting how SALoM differs in intent and implementation.

**Weaknesses**

W0: While the paper introduces several technical modules (ODEs, IB fusion, hash-based co-occurrence encoding), their individual motivations are not always clearly articulated. Clarifying why each design choice is necessary would improve the exposition.

W1: The overall writing feels verbose and sometimes disorganized, making it harder to follow the technical contributions, particularly in Sections 3.1 and 3.2.

W2: The rationale behind using Neural ODEs (and the CFC variant) for temporal memory modeling could be better justified, especially compared to simpler alternatives like attention or GRUs.

W3: Several inconsistencies and typos were found, including:
+ In Figures 2–4, the x-axis is uniformly labeled “Number of Sampled Neighbors?”, which appears to be an error.
+ Appendix B is present only with a heading but lacks any actual content.

W4: From Figure 5, performance degrades as batch size increases, potentially limiting SALoM’s scalability to larger datasets, where larger batches are often necessary for efficiency.

W5: The number of experts (Q) and selected experts (r) in the MoE mechanism are not explored in depth. Understanding their effect on performance and computational cost would be important for users aiming to apply SALoM in practice.

---

> ### Author Rebuttal · Authors · 2025-07-31
>
> We sincerely appreciate the reviewer's positive feedback on our model design and experimental rigor, as well as their constructive comments that helped improve our paper. Below we respond to the main points raised.
>
> ## W0：Individual motivations:
> We apologize for any confusion and now provide analytical and experimental motivation for each SALoM component:
>
> ### a) ODE-based memory updater for long-term temporal correlation:
> Traditional sequential methods (RNNs/GRUs) treat dynamic graphs as discrete events, struggling with temporal continuity and long-term dependencies due to gradient issues. In contrast, ODE-based methods preserve continuity and avoid gradient issues through second-order derivative integration, demonstrating superior long-term dependency capture with minimal information loss (Table 1, [1]). This experiment is designed to predict the label(-1/1) of the first node with node embedding propagated through a linear graph  of length(n).
> However, our ablation studies (Fig. 2) show that over-emphasizing long-term dependencies can cause over-globalization. We therefore propose LSMU, an MoE-based approach that dynamically balances long/short-term dependencies, adaptively selecting the optimal processing.
>
> Table 1: Accuracy under different numbers of sampling neighbors
> | |n=3|n=9|n=15|n=20|
> |---|---|---|---|---|
> |DyGFormer|100.0±0.0|53.02±6.06|42.80±16.25|42.79±19.62|
> |DyRep|100.0±0.0|47.93±2.73|48.60±2.48|50.47±2.88|
> |GraphMixer|100.0±0.0|52.80±5.56|52.49±5.36|52.04±8.20|
> |JODIE|100.0±0.0|**100.0±0.0**|60.0±14.91|50.87±2.46|
> |TGAT|100.0±0.0|47.87±2.72|50.53±2.15|49.07±1.55|
> |TGN|100.0±0.0|48.13±1.63|48.67±2.76|50.13±2.17|
> |ODE_based|**100.0±0.0**|99.93±0.21|**93.47±8.78**|**88.93±12.06**|
>
> [1]GRAVINA, Alessio, et al. Long range propagation on continuous-time dynamic graphs. *arXiv preprint arXiv:2406.02740*, 2024.
>
> ### b) GRU for short-term memory updater:
> In dynamic graphs, recent neighbor interactions typically provide the most valuable information. While simple RNNs and K-neighbor aggregation fail to effectively capture these patterns (due to vanishing gradients and limited neighbor hops respectively), GRU's gating mechanism enables robust short-term dependency learning while complementing our ODE-based long-term capture.
> Our extended experiments confirm that simple RNN suffers from gradient vanishing, leading to performance degradation (mitigated by GRU’s gating mechanism). Besides, K-neighbor aggregation is limited to 1–2 hops, as wider aggregation blurs node representations.
>
> Table 2: Trans.AP under different short-term memory updaters
> | |CanParl|USLegis|UCI|Untrade|
> |:---:|:---:|:---:|:---:|:---:|
> |RNN|98.81|89.90|96.00|92.82|
> |AggerateNeighbor|98.77|77.31|96.22|92.83|
> |GRU|**99.11**|**92.27**|**96.36**|**93.86**|
>
> ### c) MoE for long-short term fusion:
> We propose MoE to fuse long-short term temporal correlations while adaptively selecting optimal backbones and weights per input. In temporal graphs, events exhibit mixed dependencies, but naive fusion methods (e.g., concatenation, voting) treat edges uniformly, ignoring their distinct evolutionary patterns. MoE addresses this by dynamically routing events to specialized experts based on node/edge features, achieving superior performance (see Table). For instance, on USLegis, MoE improves Trans.AP by 15.3% over concatenation and 16.2% over voting, demonstrating its ability to capture varied temporal scales.
>
> Table 3: Trans.AP under different long-short term fusers
> | |CanParl|USLegis|UCI|Untrade|
> |:---:|:---:|:---:|:---:|:---:|
> |Concat|96.90|76.93|96.02|92.00|
> |Avg-Voting|98.65|76.06|96.23|92.99|
> |MoE|**99.11**|**92.27**|**96.36**|**93.86**|
>
> ### d) Co-occurrence encoder as structure encoder:
> Our co-occurrence encoder efficiently captures structural patterns while maintaining computational feasibility. Unlike traditional methods that trade off expressiveness for efficiency, it encodes neighbor co-occurrence frequencies as relative structural features, enhanced by a hash-based memory system for accelerated processing. Experiments demonstrate its superiority over basic message passing and random walk approaches, with significantly better performance at manageable computational cost.
>
> Table 4: Trans.AP with different structure encoders
> | |CanParl|USLegis|UCI|Untrade|
> |:---:|:---:|:---:|:---:|:---:|
> |MassgePassing|97.70|86.97|94.08|62.17|
> |RandomWalk|97.95|85.56|96.27|63.44|
> |Co-occurrence|**99.11**|**92.27**|**96.36**|**93.86**|
>
> ### d) IB for feature fusion:
> Temporal graphs require effective fusion of temporal and structural features, which often conflict when combined naively. Our solution adapts the Information Bottleneck method to: (1) create unified node representations, (2) perform data-aware dimensionality reduction, and (3) filter noise while preserving critical temporal-structural information. This approach outperforms naive fusion methods by resolving feature conflicts and optimizing representation quality (as justified in Fig.3).
>
> ## W1&W3&Q1&Q2: Writing issues
> We sincerely appreciate your careful review. We are currently: (1) reorganizing key sections, (2) clarifying mathematical explanations, and (3) correcting all identified inconsistencies. A thorough proofreading will ensure these issues are resolved in the final version.
>
> ## W2&Q5: Neural ODEs rationale
>
> Thanks for your insightful advice. We further justify the introduction of Neural ODEs. Unlike RNNs' discrete Backpropagation Through Time (BPTT), which triggers gradient vanishing. ODE-based methods excel in dynamic graphs by: (1) enabling continuous memory updates for irregular timestamps, and (2) preventing gradient vanishing through second-order derivative integration.
> To be specific, gradient computation in ODEs employs the adjoint method, where gradients are calculated by solving an augmented ODE backward. For instance, the adjoint state $$ a(t) = -\frac{\partial\mathscr{L}}{\partial h(t)} $$  has a time derivative given by $$ \frac{da(t)}{dt} = -a(t)^T \frac{\partial f(t, h(t), \theta_t)}{\partial h(t)} $$, ensuring gradient stability through integration.
>
> ## W4: Batch size impact
>
> Thanks for your insightful advice. We acknowledge the performance trade-off with larger batches, a common challenge for memory-based methods. Our experiments show SALoM maintains better accuracy than TGN across batch sizes (see results below), though both are affected by information loss. While training optimizations could mitigate this, our focus remains on model design.
>
> Table 5: Trans.AP comparison under different batch sizes
> |  | CanParl |  | USLegis |  | Untrade |  |
> |:---:|:---:|:---:|:---:|:---:|:---:|:---:|
> |  | TGN | SALoM | TGN | SALoM | TGN | SALoM |
> | bs=10 | 98.25 | 99.11 | 89.80 | 93.75 | 69.41 | 93.86 |
> | bs=20 | 80.95 | 97.79 | 85.32 | 87.99 | 65.24 | 90.57 |
> | bs=100 | 68.02 | 89.76 | 76.08 | 77.10 | 64.21 | 79.50 |
> | bs=200 | 61.13 | 80.13 | 74.89 | 75.91 | 60.15 | 72.84 |
> | bs=1000 | 61.87 | 72.76 | 70.40 | 71.36 | 62.88 | 65.57 |
>
>
> ## W5&Q4: MoE configuration analysis
>
> Thanks for your insightful advice. The extended experimental results are as below. Our experiments show that MoE hyperparameters (expert count Q and selection count r) generally have minimal impact on performance, with two exceptions: (1) selecting only one expert degrades performance (e.g., −15% AP on USLegis) due to lost feature fusion, and (2) excessive experts underfit small datasets (e.g., USLegis). Computational costs scale predictably: VRAM grows linearly with Q (+2.7MB/expert), while training time increases sublinearly (~0.02s → 0.027s/iter for 4→8 experts). The optimal balance is 6 experts with 2 selected, achieving peak accuracy (e.g., 93.86 AP on Untrade) with moderate resource use (18.67MB VRAM). We will incorporate these findings in the final version.
>
> Table 6: Trans.AP under different number of selected and total experts
> |n_selected|n_experts|CanParl|USLegis|UCI|Untreade|
> |:---:|:---:|:---:|:---:|:---:|:---:|
> |2|4|98.38|91.94|96.16|92.54|
> |3|4|98.92|91.09|96.42|91.54|
> |1|6|98.88|78.51|96.20|93.18|
> |2|6|99.11|92.27|96.36|93.86|
> |3|6|98.92|92.13|96.48|91.90|
> |2|8|98.70|92.13|96.51|93.19|
> |3|8|99.00|79.40|96.14|92.82|
>
> Table 7: Train time and GPU memory usage under different numbers of selected and total experts
> |n_selected|n_experts|seconds/iter|GPU memory|
> |---|---|---|---|
> |2|4|0.0213|15.98M|
> |3|4|0.0236|15.98M|
> |2|6|0.0238|18.67M|
> |3|6|0.0241|18.67M|
> |2|8|0.0276|21.37M|
> |3|8|0.0272|21.37M|
>
>
> ## Q3: Single hashmap impact
>
> Thank you for your valuable feedback. Our extended experiments confirm the limitations of using a single hashmap.
> The dual-memory structure (long-term and short-term) effectively captures both historical and recent neighbor records while reducing hash collision effects. A single hash table would degrade feature extraction, worsen collision impacts, and produce less distinguishable embeddings, ultimately harming performance. The results below support this conclusion.
>
> Table 8: Trans.AP under using SingleHash and DualHash
> |  | CanParl | USLegis | UCI | Untrade |
> |:---:|:---:|:---:|:---:|:---:|
> | Single Hash | 98.57 | 77.87 | 96.21 | 67.19 |
> | Dual Hash | **99.11** | **92.27** | **96.36** | **93.86** |

---

> > ### Comment · Area_Chair_SG7o · 2025-08-04
> >
> > Dear Reviewer, please engage into discussions with the Authors as the deadline for this key phase of the NeurIPS review process is only a couple of days away.

---

> > ### Comment · Reviewer_sYk3 · 2025-08-05
> > **Response to Authors**
> >
> > Thank you for the detailed response, and apologies for the delayed reply as I have been on a business trip over the past few days. While your response has addressed most of my concerns, I would still recommend taking advantage of the additional page allowed in the final version to further polish the writing and provide clearer explanations for some of the module motivations. This will significantly improve the readability of your work. I plan to raise my rating to 4. Good luck.

---

> > > ### Author Response · Authors · 2025-08-05
> > >
> > > Thank you for your thoughtful feedback and constructive suggestions. We sincerely appreciate the time and effort you’ve dedicated to reviewing our work.
> > >
> > > We will carefully incorporate your recommendations, particularly by refining the writing to clarify the motivations behind key modules and improving overall readability in the final version. Your guidance has been invaluable, and we truly appreciate your willingness to raise the rating.

---

### Official Review · Reviewer_64Zg · 2025-07-04

**Clarity:** 2
**Significance:** 3
**Originality:** 3
**Rating:** 4
**Confidence:** 4

**Summary:**

This paper introduces a dynamic graph learning framework called SALoM, designed to effectively capture both long- and short-range temporal dependencies while incorporating graph structural information. To overcome limitations of existing memory-based and neighbor-based methods, SALoM introduces a Long-Short Memory Updater (LSMU) that leverages a mixture-of-experts architecture to dynamically balance influences from long-term and short-term memory using neural ODEs and GRUs. It also integrates co-occurrence-based structural encoding through an information bottleneck-based fusion module, allowing temporal and structural features to be combined without conflict. Extensive experiments on multiple benchmarks demonstrate SALoM’s superior performance, particularly in dynamic link prediction tasks, where it significantly outperforms prior methods in both accuracy and adaptability.

**Questions:**

See weaknesses. Additionally:

1. What is the effect of number of experts with respect to performance?
2. Why does the performance degrade as the number of historical neighbors increase (Figure 4)?

**Ethical Concerns:**

["NO or VERY MINOR ethics concerns only"]

**Final Justification:**

The authors addressed my concerns but the information leakage issue pointed out by another reviewer is significant. I will keep my score the same.

**Limitations:**

yes

**Paper Formatting Concerns:**

Figures 2 and 3 have incorrect x-axis labels "Number of Sampled neighbors"

**Quality:**

3

**Strengths And Weaknesses:**

Strengths:

1. The idea of using mixture of experts for choosing between long and short-term memory modules in interesting and novel for dynamic graphs.
2. Significant performance improvements in the transudative setting for UN Trade and Vote datasets.

Weaknesses:

1. Weak motivation for some modules: The authors imply that GRUs are for short-term memory and CFC cells for long-term memory. Why are GRUs for short-term memory? Why not use a simpler module for short-term memory like aggregating last K neighbors as done in TGAT [1]. Rather than using an MoE module for fusion, why not concatenate the outputs of both modules? For the co-occurance encoder, why not use a simple message passing architecture instead to capture the structural information? Your architecture is relatively complicated so every module must be motivated with experiments.

2. Slight performance degradation in the inductive setting.

[1] Xu et al. Inductive Representation Learning on Temporal Graphs. ICLR. 2020

---

> ### Author Rebuttal · Authors · 2025-07-31
>
> We are glad that the reviewer appreciates our work from the perspective of novelty and performance. Thank you for pointing out the weak motivation issue. We have carefully revised the manuscript according to your constructive suggestions. Below we address the main points raised in the review.
>
>
> ## W1：Individual motivations:
> We apologize for any confusion and now provide analytical and experimental motivation for each SALoM component:
>
> ### a) ODE-based memory updater for long-term temporal correlation:
> Traditional sequential methods (RNNs/GRUs) treat dynamic graphs as discrete events, struggling with temporal continuity and long-term dependencies due to gradient issues. In contrast, ODE-based methods preserve continuity and avoid gradient issues through second-order derivative integration, demonstrating superior long-term dependency capture with minimal information loss (Table 1, [1]). This experiment is designed to predict the label(-1/1) of the first node with node embedding propagated through a linear graph  of length(n).
> However, our ablation studies (Fig. 2) show that over-emphasizing long-term dependencies can cause over-globalization. We therefore propose LSMU, an MoE-based approach that dynamically balances long/short-term dependencies, adaptively selecting the optimal processing.
>
> Table 1: Accuracy under different numbers of sampling neighbors
> | |n=3|n=9|n=15|n=20|
> |---|---|---|---|---|
> |DyGFormer|100.0±0.0|53.02±6.06|42.80±16.25|42.79±19.62|
> |DyRep|100.0±0.0|47.93±2.73|48.60±2.48|50.47±2.88|
> |GraphMixer|100.0±0.0|52.80±5.56|52.49±5.36|52.04±8.20|
> |JODIE|100.0±0.0|**100.0±0.0**|60.0±14.91|50.87±2.46|
> |TGAT|100.0±0.0|47.87±2.72|50.53±2.15|49.07±1.55|
> |TGN|100.0±0.0|48.13±1.63|48.67±2.76|50.13±2.17|
> |ODE_based|**100.0±0.0**|99.93±0.21|**93.47±8.78**|**88.93±12.06**|
>
> [1]GRAVINA, Alessio, et al. Long range propagation on continuous-time dynamic graphs. *arXiv preprint arXiv:2406.02740*, 2024.
>
> ### b) GRU for short-term memory updater:
> In dynamic graphs, recent neighbor interactions typically provide the most valuable information. While simple RNNs and K-neighbor aggregation fail to effectively capture these patterns (due to vanishing gradients and limited neighbor hops respectively), GRU's gating mechanism enables robust short-term dependency learning while complementing our ODE-based long-term capture.
> Our extended experiments confirm that simple RNN suffers from gradient vanishing, leading to performance degradation (mitigated by GRU’s gating mechanism). Besides, K-neighbor aggregation is limited to 1–2 hops, as wider aggregation blurs node representations.
>
> Table 2: Trans.AP under different short-term memory updaters
> | |CanParl|USLegis|UCI|Untrade|
> |:---:|:---:|:---:|:---:|:---:|
> |RNN|98.81|89.90|96.00|92.82|
> |AggerateNeighbor|98.77|77.31|96.22|92.83|
> |GRU|**99.11**|**92.27**|**96.36**|**93.86**|
>
> ### c) MoE for long-short term fusion:
> We propose MoE to fuse long-short term temporal correlations while adaptively selecting optimal backbones and weights per input. In temporal graphs, events exhibit mixed dependencies, but naive fusion methods (e.g., concatenation, voting) treat edges uniformly, ignoring their distinct evolutionary patterns. MoE addresses this by dynamically routing events to specialized experts based on node/edge features, achieving superior performance (see Table). For instance, on USLegis, MoE improves Trans.AP by 15.3% over concatenation and 16.2% over voting, demonstrating its ability to capture varied temporal scales.
>
> Table 3: Trans.AP under different long-short term fusers
> | |CanParl|USLegis|UCI|Untrade|
> |:---:|:---:|:---:|:---:|:---:|
> |Concat|96.90|76.93|96.02|92.00|
> |Avg-Voting|98.65|76.06|96.23|92.99|
> |MoE|**99.11**|**92.27**|**96.36**|**93.86**|
>
> ### d) Co-occurrence encoder as structure encoder:
> Our co-occurrence encoder efficiently captures structural patterns while maintaining computational feasibility. Unlike traditional methods that trade off expressiveness for efficiency, it encodes neighbor co-occurrence frequencies as relative structural features, enhanced by a hash-based memory system for accelerated processing. Experiments demonstrate its superiority over basic message passing and random walk approaches, with significantly better performance at manageable computational cost.
>
> Table 4: Trans.AP with different structure encoders
> | |CanParl|USLegis|UCI|Untrade|
> |:---:|:---:|:---:|:---:|:---:|
> |MassgePassing|97.70|86.97|94.08|62.17|
> |RandomWalk|97.95|85.56|96.27|63.44|
> |Co-occurrence|**99.11**|**92.27**|**96.36**|**93.86**|
>
> ### d) IB for feature fusion:
> Temporal graphs require effective fusion of temporal and structural features, which often conflict when combined naively. Our solution adapts the Information Bottleneck method to: (1) create unified node representations, (2) perform data-aware dimensionality reduction, and (3) filter noise while preserving critical temporal-structural information. This approach outperforms naive fusion methods by resolving feature conflicts and optimizing representation quality (as justified in Fig.3).
>
> ## W2：Performance in inductive setting
>
> Although SALoM exhibits a slight performance decline in inductive settings due to its relatively higher reliance on prior knowledge, it still achieves state-of-the-art (SOTA) performance on most datasets. Notably, it achieves an AP above 85 on the UNtrade dataset, and overall, it remains the most effective method on average across all datasets.
>
> ## Q1: Num of experts
>
> Thanks for your insightful advice, we have extended our experiments on more hyperparameter settings of LSMU module as below.
>
> For the results, we discover that in general, the hyperparameters of MoE module hold rather small impact on model performance. However, a few anomalies are discovered. First, as observed in the third row, selecting only one expert per iteration—an unconventional setting—leads to performance degradation due to the lack of fusion between long-term and short-term temporal features, resulting from selecting a single backbone for all edges in a batch. Second, the experimental setup in the last row shows poor performance on the USLegis dataset, attributed to an insufficient dataset size to meet the demands of each expert, causing all experts to be underfitted.
>
> Table 5: Trans.AP under different number of selected and total experts
> |n_selected|n_experts|CanParl|USLegis|UCI|Untreade|
> |:---:|:---:|:---:|:---:|:---:|:---:|
> |2|4|98.38|91.94|96.16|92.54|
> |3|4|98.92|91.09|96.42|91.54|
> |1|6|98.88|78.51|96.20|93.18|
> |2|6|99.11|92.27|96.36|93.86|
> |3|6|98.92|92.13|96.48|91.90|
> |2|8|98.70|92.13|96.51|93.19|
> |3|8|99.00|79.40|96.14|92.82|
>
> ## Q2: Performance degrade as the number of historical neighbors increase
>
> We appreciate this important observation. The performance degradation with increasing historical neighbors occurs because: (1) larger neighbor samples lead to higher similarity between nodes during aggregation, reducing embedding distinctiveness and potentially causing over-globalization; (2) for memory updates, iterate mechanisms already preserve long-term information, making excessive sampling counterproductive as it disproportionately weights long-term dependencies at the expense of crucial short-term patterns.

---

> > ### Comment · Area_Chair_SG7o · 2025-08-04
> >
> > Dear Reviewer, please engage into discussions with the Authors as the deadline for this key phase of the NeurIPS review process is only a couple of days away.

---

### Note · Authors · 2025-08-12

We sincerely appreciate the time and thoughtful feedback from the AC and all reviewers, which has significantly strengthened our work. We're particularly encouraged by two reviewers raising their ratings. Below, we summarize our responses and revisions.

# Individual Motivations
Reviewers [64Zg] and [sYk3] raised concerns about SALoM’s underlying motivation. In response, we further clarified our motivation both theoretically and experimentally, comparing SALoM’s each module with prior competitors, highlighting its theoretical advantages and empirical effectiveness.

# SALoM’s Design
Reviewers raised several concerns: [64Zg] and [sYk3] questioned MoE configurations and rationale for ODE-based methods, [vKih] addressed computational overhead and distinction between our MoE implementation and prior MoE-based GNNs, [dfEN] inquired about LSMU’s generalizability to other memory-based methods. In response, we: (1) extended experiments on MoE configurations and computational efficiency against competing approaches, (2) provided justification for ODE-based method, emphasizing its advantages in modeling C-TDGs and mitigating gradient vanishing, (3) clarified our diverse MoE adoption, (4) demonstrated LSMU's application to TGN.

# Evaluation Performance
Reviewer [64Zg] noted performance drops in inductive settings and with more historical neighbors, while [sYk3] and [vKih] questioned our dual-hash memory. We clarified that: (1) While performance degradation in inductive settings and larger neighbor sets can reduce embedding distinctiveness are common challenges across methods, LSMU still outperforms competing approaches. (2) The dual-hash architecture is specifically designed to capture both long- and short-term features more effectively while minimizing hash collisions.

# New Datasets and Codebases
Reviewer [dfEN] suggested additional datasets and identified a shared implementation flaw about information leakage in prior works. In response, we: (1) expanded experiments incorporating the suggested datasets, (2) rectified the common implementation flaw, and (3) further demonstrated SALoM's performance against all baselines under a fair comparison.

In summary, we have addressed nearly all key reviewer concerns through expanded experimental validation, logical analysis and implementation refinements. We deeply appreciate the reviewers' valuable feedback, which has significantly strengthened our work and will be carefully incorporated in the final version.

---

### Decision · Program_Chairs · 2025-09-17

**Decision:**

Accept (poster)

**Comment:**

The paper discusses a dynamic graph learning approach that captures both long- and short-term temporal dependencies while integrating structural information. The architecture is motivated by the need to address limitations in existing memory-based temporal graph networks and includes a mixture-of-experts mechanism to balance temporal correlations across timescales.

The initial reviews identified the work as methodologically solid, with promising performance across a broad set of benchmarks. However, a major concern was raised regarding a potential data leakage issue, stemming from improper memory updates across edges occurring at the same timestamp. This flaw, inherited from previous memory-based models, significantly impacted the trustworthiness of the reported results and risked undermining the paper’s main empirical claims.

In the post-rebuttal discussion, the authors responded promptly and thoroughly. They implemented a redesigned memory update strategy to eliminate the leakage, and they re-ran key experiments using this corrected protocol. The updated results, reported in the discussion phase, confirmed that SALoM still achieves state-of-the-art performance, even under leakage-free conditions. This strongly validates the method’s robustness and provides a valuable experimental protocol for the broader community, which has been relying on flawed practices.

The discussion was constructive and engaged, and all reviewers acknowledged the authors’ effort to address concerns with rigor and transparency. Multiple reviewers raised their scores accordingly, recognizing that the authors not only fixed the problem but improved the clarity and reproducibility of their work.

This paper delivers both a strong technical contribution and a responsible, community-beneficial correction to a prevalent benchmarking issue. The approach is well-justified, the experimental protocol is now sound, and the results are compelling.

I recommend to accept this paper.